# AHL-differential quorum sensing regulation of amino acid metabolism in *Hafnia alvei* H4

Congyang Yan,[1,2] Xue Li,[1,2] Gongliang Zhang,[1,2] Jingran Bi,[1,2] Hongshun Hao,[2] Hongman Hou[1,2]

**ABSTRACT** Quorum sensing (QS) regulation of functional metabolites is rarely reported but a common trait of some bacteria. In this study, we found that QS promoted the extracellular accumulation of glycine and serine while inhibiting the extracellular accumulation of methionine in *Hafnia alvei* H4. The correlation analysis of five QS signals with the above three QS-regulated amino acids suggested that these QS signals may have functional differences in amino acid regulation. The exogenous AHL add-back studies on genes involved in glycine, serine, and methionine metabolic pathway highlighted that *N*-octanoyl-L-homoserine lactone (C8-HSL) downregulated the expression of *sdhC*/*fumA* genes involved in the succinate to malate pathway, thereby reducing the metabolic flux of the tricarboxylic acid (TCA) cycle as an amino acid metabolism platform. Further in-depth research revealed that the QS system promoted the conversion of folate to tetrahydrofolate (THF) by positively regulating the expression of *folA* and *folM*, thus impairing the ability of folate to promote methionine accumulation. Moreover, folate positively regulated the expression of the QS signal synthesis gene *luxI*, promoting the synthesis of QS signals, which may further enhance the influence of the QS system on amino acid metabolism. These findings contribute to the understanding of amino acid metabolism regulated by QS and provide new perspectives for accurate control of metabolic regulation caused by QS.

**IMPORTANCE** As one of the important regulatory mechanisms of microorganisms, quorum sensing (QS) is involved in the regulation of various physiological activities. However, few studies on the regulation of amino acid metabolism by QS are available. This study demonstrated that the LuxI-type QS system of *Hafnia alvei* H4 was involved in the regulation of multiple amino acid metabolism, and different types of QS signals exhibited different roles in regulating amino acid metabolism. Additionally, the regulatory effects of the QS system on amino acid metabolism were investigated from two important cycles that influence the conversion of amino acids, including the TCA cycle and the folate cycle. These findings provide new ideas on the role of QS system in the regulation of amino acid metabolism in organisms.

**KEYWORDS** *Hafnia alvei*, quorum sensing, different AHLs, QS regulation, amino acid metabolism

Quorum sensing (QS) is a widespread process that bacteria use to coordinate various physiological processes with cell density through the production, release, and group-wide detection of extracellular signal molecules (1–3). Generally, the concentration of QS signal increases with the density of bacterial cells, leading to coordinated expression of various genes in the whole bacterial population (4). Most known N-acylhomoserine lactones (AHLs) used by a diverse range of Gram-negative bacteria as signal molecules have a core N-acylated homoserine-lactone ring and straight-chain fatty acyl side groups of 4–18 carbons, with varying degrees of unsaturation at certain positions

Address correspondence to Hongman Hou, houhongman@dlpu.edu.cn.

The authors declare no conflict of interest.

See the funding table on p. 16.

of the acyl side chain (5). Various AHLs have specific functions in metabolic regulation controlled by QS, for example, four types of QS signaling molecules of *Shewanella baltica* display different regulatory activities in the metabolic regulation of total volatile base nitrogen (TVB-N) (6). There has been considerable interest in the study of different types of QS signal molecules in bacteria, which can be used both to study model systems for metabolic regulation and to better understand important microbial behaviors.

The mechanism of AHL biosynthesis is an acyl transfer reaction (7), in which the LuxI, an AHL synthetase, catalyzes the transfer of acyl groups from acyl carrier proteins (acyl-ACPs) or CoA-aryl/acyl moieties to the amino groups of *S*-adenosyl-ʟ-methionine (SAM), followed by the formation of AHLs and *S*-methylthioadenosine (MTA), as shown in Fig. 1. As an important methyl supplying amino acid, methionine not only determines the biosynthesis of SAM, which, in turn, influences the synthesis of QS signal molecules (8), but also has a close relationship with other functional amino acids, especially glycine and serine (8, 9). These amino acids are major functional substances in the one-carbon pathway that influences nucleic acid, vitamin, and other physiological activities (10). Numerous studies have shown that the tetrahydrofolate (THF) cycle can directly affect the metabolic flux of glycine, serine, and methionine (11–13). Among them, homocysteine and 5-methyltetrahydrofolate (5-mTHF) generate THF and methionine under the catalysis of MetE and MetH (Fig. 1). Moreover, the key enzyme GlyA that promotes the conversion of THF to 5,10-methylenetetrahydrofolate (5,10-mTHF) also catalyzes the reaction from serine to glycine. Finally, the formed 5,10-mTHF re-enters the THF cycle through 5-mTHF. In addition, as a central hub connecting the metabolism of

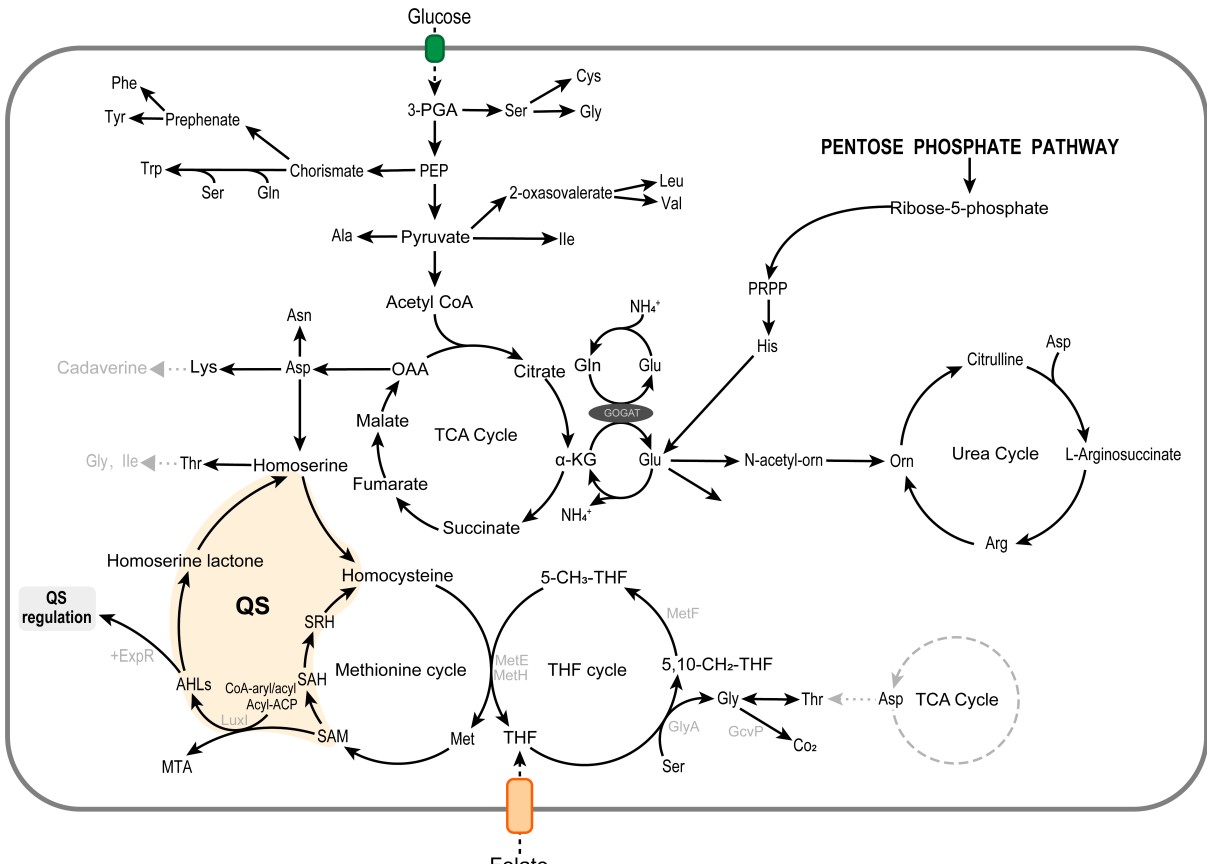

**FIG 1** Putative AHLs, amino acid, and folate metabolic network of *H. alvei* H4. AHLs production results from the conversion of SAM to MTA catalyzed by LuxI, in which SAM provides the lactone ring and acyl-ACP/CoA-aryl provides the side chains of AHLs, and then combines with homologous receptor ExpR to regulate the expression of target genes. The precursor substance of SAM biosynthesis, methionine, is closely related to the THF cycle, indicating a close relationship between the QS system, amino acid metabolism, and the THF cycle.

multiple amino acids, the tricarboxylic acid (TCA) cycle can also link these metabolic pathways of methionine, serine, and glycine (14–16). For example, succinyl-CoA is not only a component of TCA cycle but also a key precursor for the synthesis of methionine from homoserine (14). Furthermore, the metabolic pathway of serine entering the TCA cycle via pyruvate metabolism is a classic process (17–19). Zhang et al. also found that the knock-out of pyruvate kinase, which catalyzed the irreversible conversion of PEP to pyruvate, can reduce the metabolic flux of the TCA cycle, leading to a significant increase in the accumulation of serine (16). Moreover, some studies have confirmed the regulatory role of the QS system on the TCA cycle in bacteria. For example, it has been found that the QS system of *Burkholderia glumae* can upregulate the biosynthesis of glyoxalate and oxalate branching from the TCA cycle and also found that the AHLs-type QS system in *Yersinia pestis* upregulated the expression of *gltA*, *sucCD,* and *sdhCDAB* genes encoded Type II civil synthase, succinyl-COA synthase, and succinate dehydrogenase, respectively, in the TCA cycle (17). Altogether, the above studies indicated a possible link between QS and amino acid metabolism, but studies exploring this possible link were sparse. However, as far as the authors were aware, the only study that reported a link between amino acid metabolism and QS in detail was the one that amino acid-derived QS molecules can control the virulence of *Vibrio* species (20), and there are no other detailed studies that reported a link between amino acid metabolism and the QS system.

*Hafnia alvei*, a Gram-negative, motile, flagellated, facultatively anaerobic bacterium, is known to be a pathogenic and spoilage *Enterobacteriaceae* species frequently isolated from spoiled food products (21), which also possessed the AHLs-type QS system involved in various spoilage regulation, e.g., extracellular protease (22), biofilm formation (23), and bacterial motility (24). We previously investigated the potential metabolic regulatory targets of the QS system through a joint analysis of the genome and STRING database (21) and found that the QS gene cluster was closely associated with cysteine, methionine, phenylalanine, tyrosine, tryptophan, alanine, aspartic acid, and glutamate-related metabolic clusters in *H. alvei* H4. However, the regulatory mechanism by which QS mediated amino acid metabolism remains unclear. In this study, a regulatory model of QS systems responsible for specific amino acid metabolism was constructed. Key differential amino acids regulated by QS were screened and identified between *H. alvei* wild-type strain H4 and its isogenic *luxI* mutants using high-performance liquid chromatography with postcolumn fluorescence detection (HPLC-FLD) and yeast nitrogen base (YNB)-based medium, respectively. Liquid chromatography-mass spectrometry (LC-MS) was used to determine the types and metabolic rules of QS signal molecules produced by *H. alvei* H4, and then their simple linear regression with different amino acids was constructed to preliminarily determine which QS signal was responsible for the metabolic regulation of potential amino acids. Based on the influence of different AHLs-type QS systems on genes in the metabolism between different amino acids, the regulation mechanism of QS-regulated amino acids was preliminarily investigated, which can further broaden the research scope of QS regulation and enable more spoilage process caused by QS to be accurately controlled in microorganisms.

## RESULTS

### Influence of QS system on the metabolism of 21 amino acids

Our previous research found that QS gene clusters were closely connected to amino acid gene clusters through a combined analysis of the genome and STRING database (21). To identify amino acid regulated by the QS system, the *H. alvei* H4 wild-type strain (WT), its isogenic *luxI* mutant (Δ*luxI*), and *luxI* complementary strain (C-Δ*luxI*) were cultured in LB medium for 12 h, and the content of amino acids in the supernatant fractions of these strains was detected at an excitation wavelength of 293 nm (Fig. 2A). By comparing the peak areas of the three strains, it was preliminarily determined that the QS system led to metabolic differences of eight amino acids, which were L-serine, L-glycine, L-alanine, L-proline, L-methionine, L-phenylalanine, L-tryptophan, and L-lysine according to the

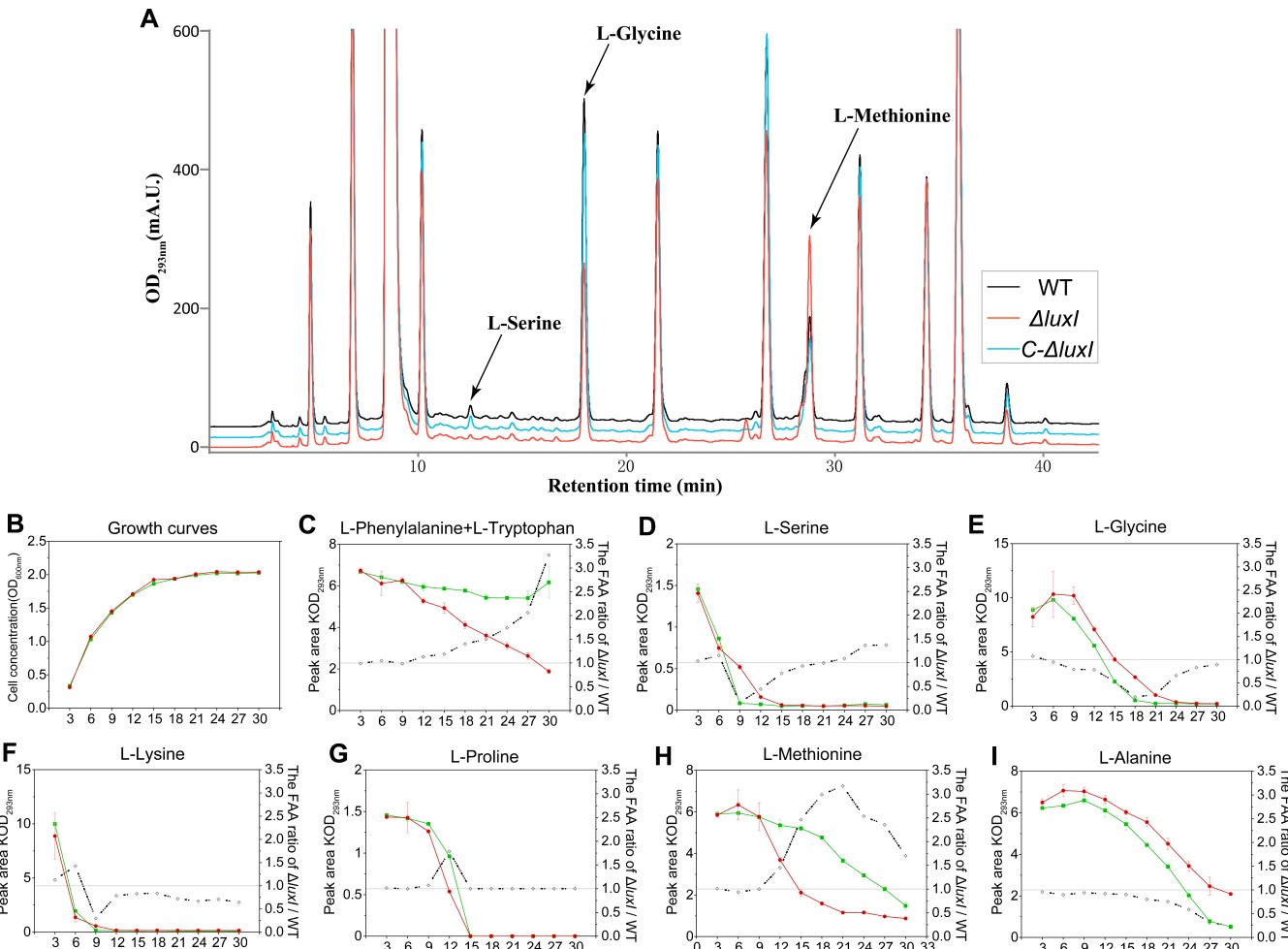

**FIG 2** Estimated effect of QS system on amino acid metabolism in *H. alvei* H4. HPLC-FLD analysis of the amino acid metabolism of WT (red line), Δ*luxI* (green line), and C-Δ*luxI* (blue line) (A). The growth curves of the above three strains in LB medium (B). Metabolic analysis of 8 amino acids regulated by the QS system in three strains, including serine (C), glycine (D), alanine (E), proline (F), methionine (G), phenylalanine/tryptophan (H), and lysine (I). The amino acids ratio of Δ*luxI*/WT (dotted line) was determined, where the ratio higher and lower than 1 represented the up- and downregulation of amino acid catabolism by QS, respectively. Data in C-I represent the mean values ± SD (three biological replicates). Statistics were achieved by one-way ANOVA: *$P < 0.05$; **$P < 0.01$; ***$P < 0.001$.

retention time sequence. Before further analyzing the amino acid metabolic trends of the three strains, the growth of these three strains in LB medium was compared. The results showed that these strains grew at similar rates, indicating that the deletion of *luxI* did not affect the growth of this strain in LB medium (Fig. 2B) and that QS system did not interfere with the subsequent analysis of amino acid difference by affecting biomass. Subsequently, the trend analysis of amino acid metabolism among these three strains revealed that the QS system of *H. alvei* H4 promoted the extracellular accumulation of four amino acids, namely, L-serine (Fig. 2C), L-glycine (Fig. 2D), L-alanine (Fig. 2E), and L-lysine (Fig. 2F). On the contrary, the QS system reduced the extracellular accumulation of other four QS-regulated amino acids, namely, L-methionine (Fig. 2G), L-phenylalanine/L-tryptophan (Fig. 2H), and L-proline (Fig. 2I). Interestingly, the differences between WT and Δ*luxI* in concentrations for these amino acids all took place between the 6 and 9 h timepoints of the experiment. At these timepoints, the cell growth was in the middle and late exponential stages of high-density population, indicating that the difference of these amino acids was more likely to be the regulatory result of density-dependent regulation mechanism-QS. Thus, the contribution of QS to regulating these amino acids needs further confirmation.

## Further screening and confirmation of amino acids regulated by QS

To further evaluate whether QS system was required for the regulation of the above differential amino acids metabolism, spot assays of *H. alvei* H4 WT strain and Δ*luxI* were performed on minimal media with or without each of the above eight amino acids as the organic nitrogen source. If there was a difference between the growth of these two strains, it indicated that the QS system was involved in the metabolic utilization of this amino acid. YNB medium and YNB supplemented with aspartate (no differential amino acids regulated by QS) served as growth controls. *H. alvei* H4 WT strains were grown on amino acids-free minimal medium supplemented with each of L-phenylalanine, L-tryptophan, L-serine, L-glycine, L-alanine, L-methionine, L-lysine, L-proline, and L-aspartate, and it was shown that they were to utilize all of these amino acids. However, the growth of Δ*luxI* on solid YNB medium with each of L-serine, L-methionine, and L-glycine was significantly different from that of the WT strain. Among them, adding L-serine or L-methionine to YNB medium significantly inhibited the growth of the Δ*luxI*, while the growth of Δ*luxI* was better than that of the WT strain when the YNB medium supplemented with L-glycine (Fig. 3). Therefore, these growth assays suggested that the QS system participated in the regulation of these three amino acid metabolisms, which was finally manifested through growth differences. The three amino acids (L-serine, L-methionine, and L-glycine) that cause differences in the growth of WT strains and Δ*luxI* may be directly regulated by the QS system, while the metabolic differences of other five amino acids may be the result of indirect regulation by QS system in *H. alvei* H4.

## Detection and identification of AHLs and their relationship with glycine, serine, and methionine

To investigate the roles of various QS signals in regulating amino acid metabolism, we first determined how many different QS signaling pathways exist in *H. alvei* H4. In addition to the three signal molecules (C4-HSL, C6-HSL, and 3OC8-HSL) identified in our previous study (25), we found that *H. alvei* H4 produced another two AHL-type signal molecules, C8-HSL and 3OC6-HSL, by unbiased metabolomics. Together, there were five QS signal pathways in *H. alvei* H4 (Fig. 4A and B). UPLC-MS was performed to determine the metabolic trends of five QS signals in the supernatant of WT strains cultured in LB medium for 30 h (Fig. 4C). The results revealed five distinct *N*-acyl-homoserine lactones-containing components in the culture supernatant extracts of *H. alvei* H4 (Fig. 4D). The five types of AHL compounds were identified as C4-HSL ($m/z$ 172.1 → 102), C6-HSL ($m/z$ 200.1 → 102), C8-HSL ($m/z$ 228.1 → 102), 3OC6-HSL ($m/z$ 214.1 → 102), and 3OC8-HSL ($m/z$ 242.1 → 102). The metabolic trends of five QS signals are shown in Fig. 4D indicated that C6-HSLwas the major AHL, with a signal of >2-fold higher than that of the other signals when incubated for more than 15 h. Specifically, long-chain AHLs (C8-HSL) were

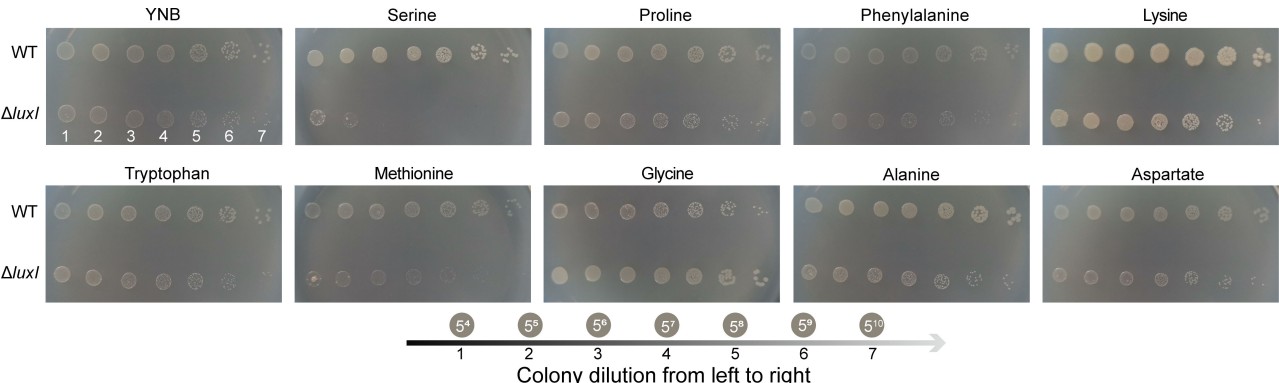

**FIG 3** Screening of amino acids directly regulated by QS. The inoculum and serial 1/5 dilutions of the *H. alvei* H4 grown overnight in LB were inoculated onto minimal media supplemented with 10 mM concentration of the indicated amino acid. Plates were photographed after 48 h, and the growth of Δ*luxI* was compared to that of WT strain.

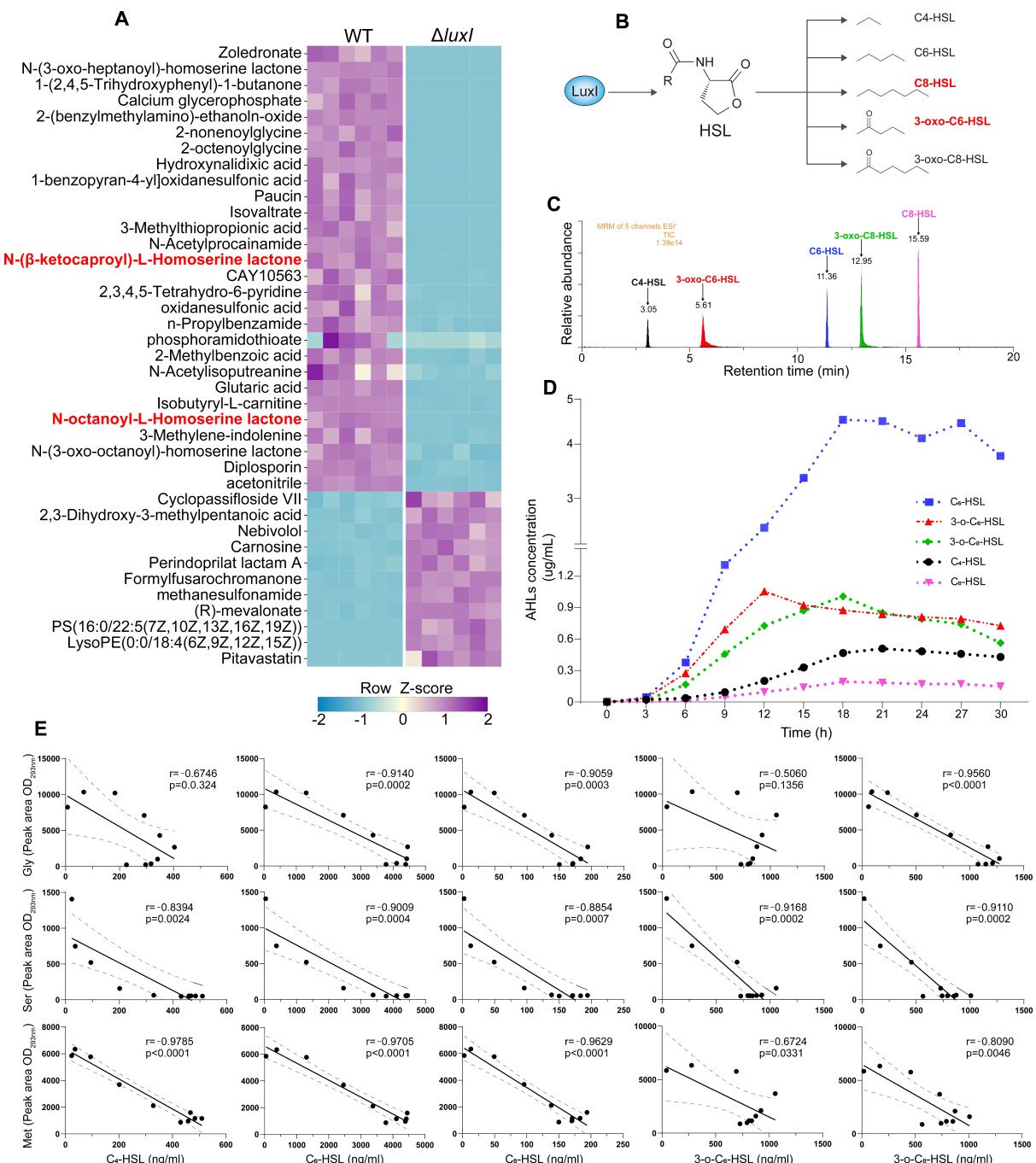

**FIG 4** Mass spectrometric analyses of AHLs produced by *H. alvei* H4 and their relationship with three different amino acids. Newly discovered two AHLs identified through unbiased metabolomics between WT and Δ*luxI* grown in LB medium for 12 h (A). The chemical structures of five AHLs signal molecules (B), three of which were found in our previous studies. Total ion current (TIC) UPLC-MS/MS chromatograms of the five synthetic AHLs standards with a concentration of 100 ng/mL (C). Broken lines illustrating the metabolic rule of above five signals produced by *H. alvei* H4 incubated in LB within 30 h (D). Correlations between the five QS signals and three differential amino acids (E). Among them, the Pearson's *r* of glycine with C4-HSL, C6-HSL, C8-HSL, 3OC6-HSL, and 3OC8-HSL were −0.6746, −0.9140, −0.9059, −0.5060, and −9,560, respectively; the Pearson's *r* of serine with C4-HSL, C6-HSL, C8-HSL, 3OC6-HSL, and 3OC8-HSL were −0.8394, −0.9009, −0.8854, −0.9168, and −9,110, respectively; the Pearson's *r* of methionine with C4-HSL, C6-HSL, C8-HSL, 3OC6-HSL, and 3OC8-HSL were −0.9785, −0.9705, −0.9629, −0.6724, and −8,090, respectively.

less abundant but detectable. In addition, we constructed a simple linear regression analysis between these five QS signals and the above three QS-regulated amino acids at the same timepoint in the supernatant of WT strains to preliminarily judge the possible

regulatory role of each QS signals in these amino acid metabolisms. It can be seen from Fig. 4E that levels of C6-, C8-, and 3OC8-HSL type QS signals were strongly correlated with glycine; C6-, 3OC6-, and 3OC8-HSL type QS signals were closely related to serine; and C4-, C6-, and C8-HSL type QS signals were closely related to methionine during the process of LB culture for 30 h. These $r$ values listed above were approximately 0.91, except for those that were less than 0.88. However, the Pearson's $r$ between aspartate and these five QS signals were all below 0.88 (Fig. S1), which could further enhance the reliability of the aforementioned analysis of QS regulation on the three types of amino acids.

## Regulation of metabolism among glycine, serine, and methionine by five QS signal molecules

To investigate the regulatory role of five QS signals in the metabolism of glycine, serine, and methionine, the potential target genes of QS regulating amino acid metabolism were obtained based on genome and transcriptome analysis. First, we found two genes, *metK* and *mtnN* encoded the methodine adenosyltransfer and *S*-adenosylhomocysteine nucleosidase respectively, directly associated with the QS gene cluster through protein-protein interaction network (PPIN) of the QS gene cluster and the amino acid metabolic gene clusters obtained from genome KEGG annotation (Fig. S2). Among them, *metK* was also associated with other five genes (including *metE*, *msrC*, *mgl*, *metH*, and *tyrB*), while *mtnN* was directly linked to *luxS* gene alone, an interspecific QS regulatory unit, and then indirectly affected the expression of more amino acid metabolism genes through this pathway. In addition, 954 differential genes were obtained from the transcriptome analysis between the WT and Δ*luxI* grown in LB medium for 12 h, of which QS upregulated and downregulated 460 and 494 genes, respectively (Fig. S3A). Among them, the number of genes involved in amino acid metabolism as well as with $\log_2$ FC below −1 and above 1 accounted for 6.5% of 954 differential genes between WT and Δ*luxI* (Fig. S3B), a total of 62, of which 25 genes were significantly upregulated and 37 genes were downregulated (Fig. S3C). These results suggest that the QS system may promote the accumulation of amino acids by downregulating the gene necessary for amino acid metabolism.

These target genes found by genome and transcriptome analysis were matched to the metabolic pathways related to three different amino acids as much as possible, which were then subdivided into four metabolic pathways: glycolysis to serine, the metabolism between glycine, serine, methionine, glycine to tricarboxylic acid cycle (TCA), and amino acid ABC transport system (Fig. 5A). After the five QS signals were added back to the culture medium of Δ*luxI* for 12 h, the recovery levels of these signals on the expression of above pathway genes were in the order of *sdhC*, *fumA,* and *metH*, involved in the synthesis of successive dehydrogenase, fumarate hydratase, and methyl synthase, respectively (Fig. 5B). The expressions of these three genes in WT strains relative to Δ*luxI* were −3-fold, −4-fold, and +1.6 fold, respectively. To further study the regulation of five QS signals on these three genes, we studied the expression of *sdhC*, *fumA,* and *metH* in WT strain and Δ*luxI* cultured in the LB medium with or without exogenous signaling molecules at four time points of the bacterial growth curve (9, 12, 15, and 18 h) and found that the expression of these genes in target strains began to show significant differences at 12 h, indicating that these genes were affected by a density-dependent regulatory mechanism (Fig. 5C). It was also found that the C8-type QS signal pathway can significantly restore the expression of *sdhC* and *fumA* genes, while C6-type QS signal pathway was more responsible for the regulation of *metH*. The above results indicated that the QS system of *H. alvei* H4 may use a C8-type signal pathway to reduce the metabolic flux of the TCA cycle as a hub for amino acid metabolism (26), leading to the difference of three amino acids, but this needs further investigation. Based on these results, we investigated the effect of different concentrations of targeted QS signals on the expression of corresponding genes (Fig. 5D). The upward trend of *metH* and the downward trend of the other two genes (*sdhC* and *fumA*) were consistent

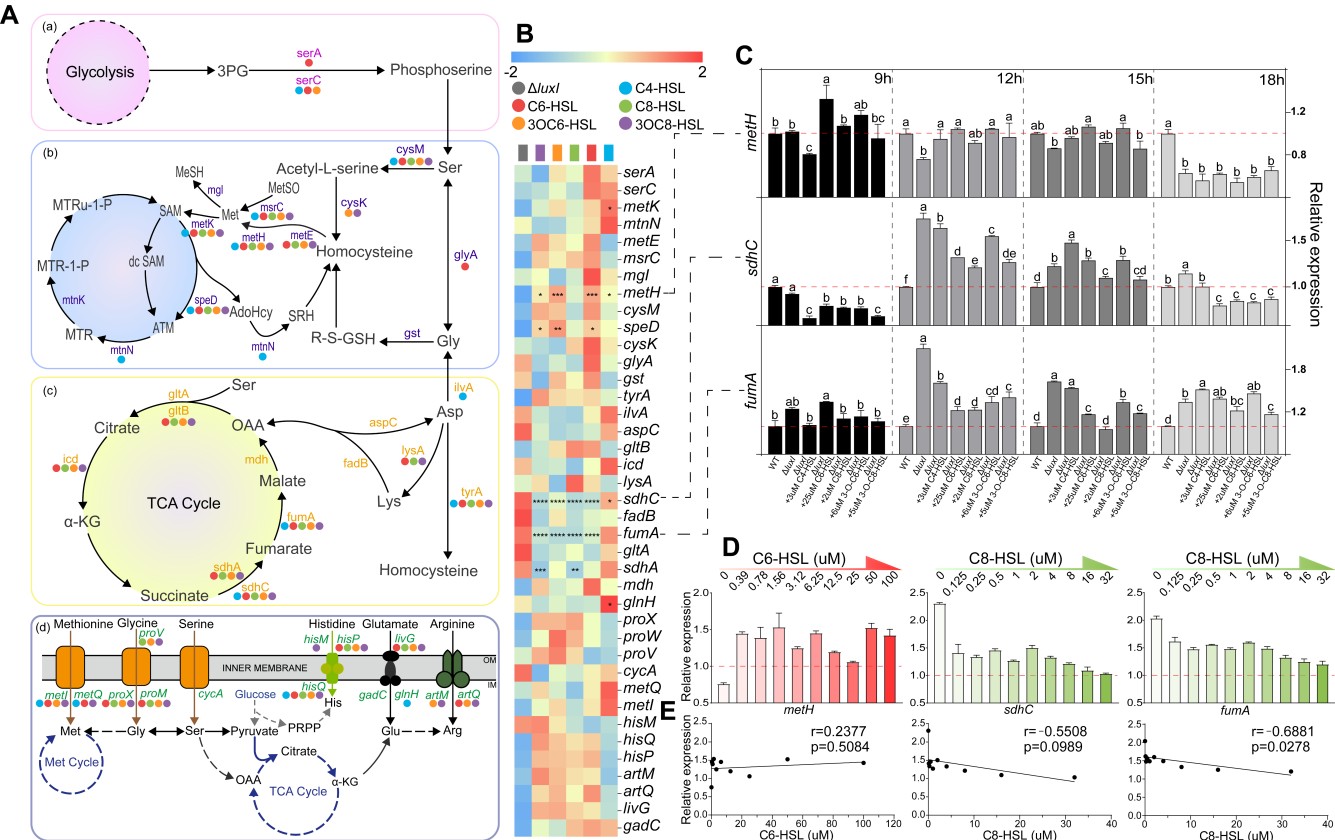

**FIG 5** Effects of five QS signaling molecules on the metabolism of glycine, serine, and methionine. Metabolic pathway genes of glycine, serine, and methionine regulated by five QS signaling molecules (A), and their regulatory ability (B). The regulatory tendency of five signal molecules on the top three genes according to the regulatory results of signal to metabolic pathway genes (C). Regulation of corresponding genes by selected signal molecules with different concentrations (D) and correlation between signal concentrations and gene expression (E).

with the results in Fig. 5G. However, the expression of these genes does not have a linear relationship with the concentration gradient of targeted QS signals (Fig. 5E). This indicated that they were still influenced by other factors, which also required further research.

## QS influenced the extracellular accumulation of glycine, serine, and methionine by inhibiting folate biosynthesis

Based on the fact that the folate cycle is an important metabolic pathway connecting glycine, serine, and methionine (11, 13, 27–29), it was necessary to further investigate the effect of folate on the metabolism of these three amino acids. First, we analyzed the regulatory effects of the QS system in *H. alvei* H4 on the folate metabolism (Fig. 6A), THF (Fig. 6B), and 5-mTHF (Fig. 6C). These results showed that the abundance of folate and 5-mTHF in WT strains were lower than that of Δ*luxI* strains, whereas THF was more abundant in WT strains. In addition, we also analyzed the effect of different concentrations of folate on the metabolism of these three differential amino acids regulated by QS. As shown in Fig. 6D through F, the addition of folate had no significant effect on the metabolism of glycine and serine, except that the addition of 12.5 and 200 µM folate significantly promoted the extracellular accumulation of glycine and serine, respectively. However, the addition of all concentrations of folate significantly promoted the extracellular accumulation of methionine. Furthermore, by exploring the effects of folate on the metabolism of glycine, serine, and methionine in the Δ*luxI* strain, it was found that the effects of folate on these three amino acids in the Δ*luxI* strain were weaker than that of WT strains (Fig. S4), which provided evidence that folate could

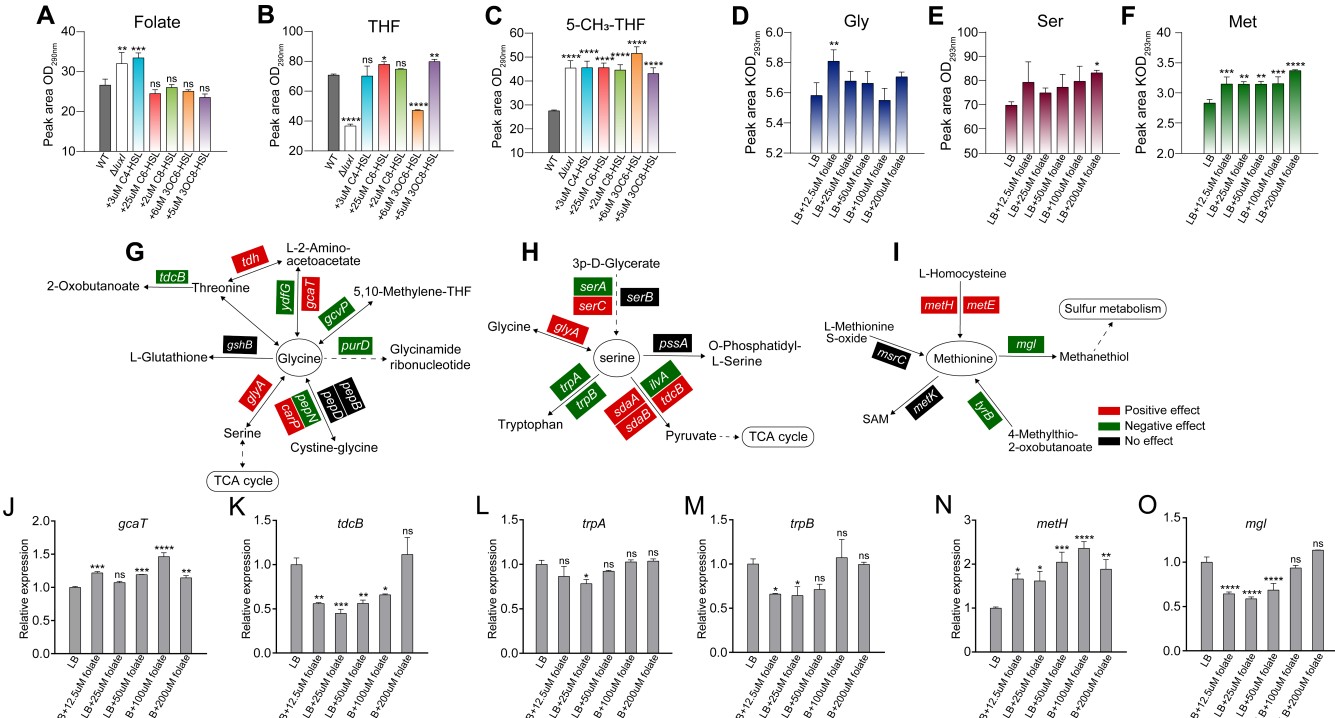

**FIG 6** The regulatory effects of QS on folate metabolism and its impact on glycine, serine, and methionine metabolism. The metabolic recovery ability of five exogenous signal molecules to folate (A), THF (B), and 5-mTHF (C) in Δ*luxI* compared to WT strain. The impact of different concentrations of folate on the metabolism of glycine (D), serine (E), and methionine (F). The regulatory effects of adding folate to LB medium on the metabolic pathway genes of glycine (G), serine (H), and methionine (I). Red represents positive effect; green represents negative effect; black represents no effect. The effects of different concentrations of folate on the expression of glycine metabolic pathway genes *gcaT* (J) and *tdcB* (K), serine metabolic pathway genes trpA (L) and trpB (M), and methionine metabolic pathway genes *metH* (N) and *mgl* (O).

enhance the regulatory role of the QS system in amino acid metabolism. Subsequently, we investigated the regulatory effects of different concentrations of folate on the expression of genes related to the metabolism of these three amino acids in *H. alvei* H4. The results showed that the effect of folate on the metabolism of these three differential amino acids was complex (Fig. 6G through I). Two key genes most significantly affected by folate were selected from each amino acid metabolism pathway for analysis. Among them, *gcaT* encoding glycine C-acetyltransferase and *tdcB* encoding catabolic threonine dehydratase (Fig. 6J and K) were, respectively, promoted and inhibited by folate, which meant that folate may promote the extracellular accumulation of glycine by influencing the metabolism between glycine and threonine. In serine metabolism, folate mainly inhibited the expression of *trpA* and *trpB* genes involved in the conversion of serine to tryptophan. This reduced the conversion of serine to tryptophan, thereby promoting the accumulation of serine (Fig. 6L and M). In the methionine metabolism, the addition of 100 μM folate resulted in a 5.13-fold upregulation of the transcriptional expression of the *metH* gene encoding the cobalamin-dependent methionine synthase, relative to the group without folate supplementation (Fig. 6N). Moreover, the addition of 25 μM folate led to a 3.25-fold downregulation of the transcriptional expression of the *mgl* gene encoding the methionine gamma-lyase, compared to the group without folate supplementation (Fig. 6O). These combined effects may promote the extracellular accumulation of methionine. The above results indicated that folate promoted the accumulation of glycine, serine, and methionine, especially methionine. Hence, QS system could weaken the biosynthesis of methionine by inhibiting folate biosynthesis, which partially explains why the WT strain has lower extracellular methionine content compared to the Δ*luxI* strain.

## Analysis of the relationship between the QS system and folate metabolism

To elucidate the relationship between the QS system and folate metabolism, we first examined the influence of the QS system on genes involved in folate metabolic pathways (Fig. 7A). Interestingly, it was observed that loss of *luxI* gene led to reduced expression of all genes involved in the folate cycle pathways (Fig. 7B). Specifically, the QS system upregulated the expression of folate catabolism genes *folA* and *folM*, thereby reducing folate accumulation and promoting the biosynthesis of THF. Although the QS system also promoted the expression of the gene *gcvT* involved in THF catabolism, it tends to upregulate the expression of THF anabolism pathway genes, leading to THF accumulation. Furthermore, we analyzed the impact of folate on the QS system in *H. alvei* H4 and found that adding 100 µM folate to the LB medium significantly promoted the production of purple pigment of CV026 strain (Fig. 7C), indicating that folate can enhance the production of QS signals in *H. alvei* H4. The impact of different concentrations of folate (12.5–200 µM) on the QS pathway genes showed that folate upregulated the expression of the gene *luxI* encoding QS signal synthesis protein (Fig. 7D). However, low concentration (12.5 µM) and high concentration (50 µM) of folate

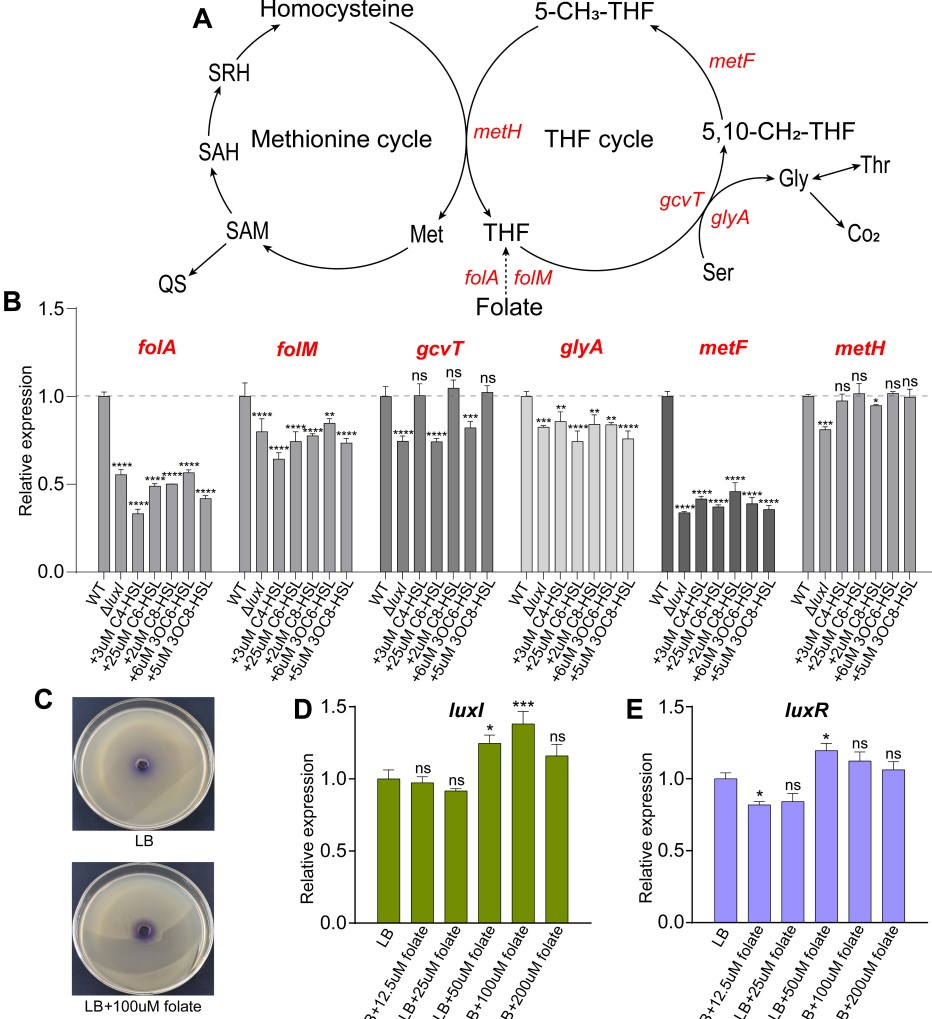

**FIG 7** Analysis of the relationship between the QS system and folate metabolism. The folate metabolism pathway of *H. alvei* H4 (A). Regulatory effects of the QS system on genes involved in the folate metabolism pathway (B). The impact of adding folate to LB medium on the production of QS signals in *H. alvei* H4 (C). The regulatory effects of different concentrations of folate on the QS system genes *luxI* (D) and *luxR* (E), encoding QS signal synthesis proteins and QS signal receptor proteins, respectively.

suppressed and promoted the expression of gene *luxR* encoding the QS signal receptor protein, respectively (Fig. 7E). In summary, the complex relationship between the QS system and folate metabolism, including the inhibitory effect of the QS system on folate accumulation and the promoting effect of folate on the synthesis of QS signals, was one of the key factors influencing the regulation of QS-mediated amino acid metabolism.

## DISCUSSION

As one of the important regulatory units of microorganisms, QS participates in the regulation of various physiological activities, including food spoilage (6, 30), metabolism (31), and virulence (3, 32–34). Studies on QS regulatory function usually focus on its effect on virulence/pathogenesis while ignoring its role in regulating metabolites. Although some studies in recent years have reported in detail that QS can regulate some functional metabolites, such as nitrogen oxide(s) and pyruvate (35, 36), there are still relatively few studies in this field compared with QS in virulence/pathogenicity. This study aimed at filling this gap by elucidating the relationship between the QS system and amino acid metabolism in the food-borne spoilage bacteria *H. alvei* H4. We speculated that there may be a certain relationship between QS and amino acid metabolism based on the following two points: first, our previous study found that QS gene clusters were closely related to amino acid-related gene clusters through the joint analysis of the genome and STRING database (21); Second, the QS signal synthesis pathway itself requires the participation of methionine, an important functional amino acid (31). In order to verify the above speculation, *H. alvei* H4 was taken as the target strain to study the regulatory mechanism of its AHLs-type QS system on amino acid metabolism in LB culture system.

Amino acids are not only important but essential basic nutrients in life activities, so their metabolic transformation and regulation have received extensive attention (37). Several lines of evidence suggested that QS may impact amino acid metabolism in microorganisms. For example, Kang et al. showed that the QS system in *Burkholderia glumae* decreased glutamate uptake by downregulating the expression of glutamate transporter genes (38). In addition, Davenport et al. found that *Pseudomonas aeruginosa* using AHL-type QS systems can inhibit the metabolism of alanine (39). Interestingly, different QS systems in these studies almost all inhibit the consumption of nutrients, which was consistent with our research results on QS inhibiting the metabolism of glycine and serine. Furthermore, our transcriptomic results between *H. alvei* WT strain and Δ*luxI* showed that there were 62 genes related to amino acid metabolism, of which the number of genes upregulated by QS was significantly less than the number of genes downregulated. This indicated that QS could downregulate the expression of amino acid metabolism-related genes at the transcriptional level, thereby avoiding nutrient depletion and the production of toxic substances. This conclusion was consistent with a report demonstrating QS acts as a metabolic brake on individuals when cells begin to mass, implying a mechanism by which AHL-mediated QS might have evolved to ensure homeostasis of the primary metabolism of individuals under crowded conditions (17).

In addition to methionine normally involved in AHLs synthesis, other amino acids regulated by the QS system may also directly participate in the biosynthesis of QS signal molecules and then regulate the physiological activities of cells. For example, Defoirdt reported that some amino acid-derived signal molecules, including indole, cyclo(L-phenylalanine-L-proline) (cyclo(Phe-Pro)), and 3,5-dimethylpirazin-2-ol (DPO), can control the virulence of *Vibrios* (20). The rich QS signal types provide the necessary conditions for the diversity of QS regulation, that is, different types of QS signal molecules have their preferences when regulating the expression of target genes. As the previous study by Kang et al., only C8-type QS signal molecules can restore the glutamate in *tofI*::Ω strains to the level of WT (38) although *B. glumae* can produce other types of QS signal molecules (40). Consistent with the above research conclusions, adding five AHLs-type QS signals determined by LC-MS back to the Δ*luxI* fermentation broth showed that different types of QS signals also exhibited their respective regulatory

characteristics whether they restored the genes on the metabolic pathway of glycine, serine, and methionine, or genes on the THF cycle pathway, as shown in Fig. 5B and 7B. Subsequently, this conclusion was confirmed by the qRT-PCR experiments of different times and concentrations of the same QS signal on the most significant restoring genes, in which C6- and C8-type QS signal pathways were determined to be responsible for regulating the expression of *metH* gene in methionine metabolism pathway and *sdhC/fumA* genes in TCA cycle pathway, respectively.

The TCA cycle provides a platform for the transformation and connection between various nutrients of microorganisms, including sugars, nucleic acids, amino acids, etc. (26). The mutual transformation and metabolic homeostasis of amino acids are also closely related to the TCA cycle, in which aspartate is one of the important communication channels between amino acid metabolism and the TCA cycle (26). It was found that the metabolism of different amino acids (glycine, serine, and methionine) regulated by QS can be linked with aspartate in the same pathway, so the metabolic flux of TCA cycle regulated by QS may also contribute to the difference of these amino acids. A series of studies have confirmed that the QS system can participate in the regulation of TCA cycle. For example, *citA* and *gltA* encoding citrate synthase in *Staphylococcus* and *Yersinia pestis*, respectively, were upregulated by QS (17). Other genes of the TCA cycle, including *aceB* and *aceA* (encoding malate synthase A) were also upregulated by QS in *Yersinia pestis* (17). However, we found that our results were inconsistent with the above studies, namely, QS downregulated the transcriptional expression of *sdhC* and *fumA* (encoding successful dehydrogenase and fumarate hydrase, respectively). As far as we know, these two genes of TCA cycle have not been previously reported to be regulated by QS, which further enriches the content of QS regulating the TCA cycle. These data indicated that TCA cycle pathway genes may be regulated by QS in both positive and negative directions. However, the biological significance of QS-mediated TCA cycle control remains unclear, which requires further research.

Although the TCA cycle provides an important platform for the metabolism of amino acids, there is still a limited understanding of how the QS system specifically regulates the metabolism of glycine, serine, and methionine. Given that the folate cycle provides a crucial link connecting these three amino acids (29), it was necessary to further explore their interrelationships. This study found that the addition of different concentrations of folate promoted the biosynthesis of methionine, which was consistent with most studies also reported that folate can enhance methionine biosynthesis (11, 13, 27, 41). However, Acho´n et al. reported that folate had no effect on the accumulation of methionine (42). In addition, contrary to the findings of this study showing that folate promoted glycine and serine biosynthesis, Maloney et al. reported that folate reduced the biosynthesis of glycine and serine (27). This study also found that the QS system inhibited folate biosynthesis, which was supported by the work of Wang et al. showing that QS system reduced folate accumulation by inhibiting the expression of folate biosynthesis pathway genes (43). The inhibition of folate by QS may indirectly decrease methionine biosynthesis, partially explaining why the WT strain produced lower levels of methionine compared to the Δ*luxI*. However, the above findings still cannot explain the results of QS system promoting the extracellular accumulation of glycine and serine from the perspective of QS inhibiting folate metabolism, which needs further study. Furthermore, this study found that folate significantly upregulated the expression of the gene *luxI* encoding AHLs synthesis protein and promoted the accumulation of methionine, which together contribute to the production of QS signals. However, the QS system promotes the consumption of folate and the synthesis of THF, thereby accelerating the THF cycle. This situation should have promoted the conversion of homocysteine to methionine, leading to the accumulation of methionine. However, the QS system reduced the accumulation of methionine instead. These results suggested that the QS system was more inclined to promote the consumption of methionine, which was consistent with the positive regulation of *metK* by the QS system, ultimately enhancing the synthesis of QS signals. Kopp et al. reported that folate promoted the formation of SAM (28), which

may stimulate the expression of the gene *luxl*. In conclusion, the microbial QS system has a complex relationship with microbial metabolism. Deciphering the metabolic regulation involved by different type of QS signals will help to achieve a more detailed microbial metabolic control.

## MATERIALS AND METHODS

### Bacterial strains and culture conditions

The *H. alvei* WT, Δ*luxl*, and C-Δ*luxl* were routinely grown in LB medium (Luria Bertani, 10 g/L tryptone, 5 g/L yeast extract, 10 g/L NaCl). When solid medium was required, 15 g/L agar was added. Cells from overnight cultures were washed twice with phosphate-buffered saline (PBS), counted, and inoculated to the experimental medium. For spot assay, cells were spotted on minimal agar media (2% glucose, 0.17% YNB [without amino acids], 1.5% agar) containing indicated concentrations of targeted nitrogen sources. Among them, 100 g of YNB medium included 5.0 g ammonium sulfate, 2.0 µg biotin, 400.0 µg calcium pantothenate, 2.0 µg folate, 2000.0 µg inositol, 400.0 µg niacin, 200.0 µg *p*-aminobenzoic acid, 400.0 µg pyridoxine hydrochloride, 200.0 µg riboflavin, 400.0 µg thiamine hydrochloride, 500.0 µg boric acid, 40.0 µg copper sulfate, 100.0 µg potassium iode, 200.0 µg ferric chloride, 400.0 µg manganese sulfate, 200.0 µg sodium molybdate, 400.0 µg zinc sulfate, 1.0 g monopotassium photsphate, 0.5 g magnesium sulfate, 0.1 g sodium chloride, and 0.1 g calcium chloride. Solutions of nitrogen sources were subjected to sterile filtration and added to the media after autoclaving.

### Amino acid analysis

The *H. alvei* WT and Δ*luxl* were grown overnight in 100 mL LB. Cultures were inoculated to an optical density at 600 nm ($OD_{600}$) of 0.15 and grown for 12 h. One milliliter of media was collected every 3 h and pelleted for 10 min at 8,000 rpm. Supernatants were collected and derivatization with Dansyl chloride (DCl). The reaction mixture was prepared in 1.5 mL Eppendorf Safe Lock Tubes and consisted of 70 µL of sample, 100 µL of DCl, and 0.2 M $Na_2B_4O_7 \cdot 10H_2O$ (pH 9.3) solution up to a final volume of 1,000 µL. The mixture was incubated for 30 min at 40°C in an ultrasonic bath and then centrifuged at 12,000 rpm for 10 min. The supernatants were recovered and diluted with MeOH (1:1, vol/vol) for HPLC-FLD analysis.

### Spot assays

Exponentially growing cells were harvested from LB at 30°C, washed twice with PBS, and adjusted to an $OD_{600}$ 1.7. The cells were serially diluted, and 8 µL was spotted onto the minimal agar media with 1.5% agar. Plates were incubated at 30°C, checked at regular intervals, and scanned after 48 h.

### Protein-protein interaction network (PPIN)

The PPIN of QS and amino acid-related genes in *H. alvei* H4 was retrieved from the STRING database (a database of known and predicted protein-protein interactions commonly used for mining core regulatory genes and functional proteins; http://string-db.org/) based on the KEGG annotation of the genome. The resulting PPIN network was visualized with Cytoscape (version 3.8.2, http://www.cytoscape.org), and the size of the node and the transparency of the edge were used to represent the MCODE Score.

### RNA-seq

RNA of *H. alvei* WT and Δ*luxl* grown in LB medium for 12 h were extracted with a RNAprep pure Cell/Bacteria Kit (Tiangen Biotech, Beijing, China) according to the

manufacturer's instructions. Each group consisted of three replicates. RNA was quantified using a Qubit 2.0 Fluorometer (Life Technologies, CA, USA). Library preparation was performed using NEBNext Ultra Directional RNA Library Prep Kit for Illumina (NEB, USA) and was quality assessed using an Agilent Bioanalyzer 2100 system. Libraries were pooled and sequenced using the Illumina Hiseq platform. Differential expression analysis was performed using the DESeq R package (1.18.0). DEseq was based on a model using a negative binomial distribution to estimate the variance-mean dependence and differential expression of the count data from high-throughput sequencing analysis. Genes with a $\log_2$ FC value above 1 or below −1 and with an adjusted $P$ value of <0.05 were designated for differential expression.

## Unbiased metabolomic fingerprint by LC-MS/MS

The WT strain and Δ*luxI* were cultured in LB medium at 30°C. The supernatants were harvested at 12 h and quickly stored at −80°C prior to lyophilization. Each group consisted of six replicates. The powder of freeze-dried samples was soaked in an extraction solvent containing methanol/acetonitrile (2:1, by vol), with 0.3 mg/mL 2-chloro-L-phenylalanine used as an internal standard. All extracts were vortexed for 30 s and centrifuged for 15 min (13,000 rpm at 4°C). The metabolite analysis was performed on an ACQUITY UPLC I-class system (Waters Corporation, Milford, USA) coupled with VION IMS QTOF mass spectrometer (Waters Corporation, Milford, USA) with an electrospray ionization source, and an ACQUITYUPLC BEH C18 column (1.7 µm, 2.1 × 100 mm).

## Extraction and detection of AHLs

The extraction of AHLs from the supernatants of *H. alvei* was conducted as described by reference (2), with slight modification. AHLs were extracted trice from 80 mL cell-free supernatant with acidified ethyl acetate (0.1%, vol/vol, formic acid). The extracts were removed by rotary evaporation, and the residue was resuspended in 1 mL of methanol. AHLs in the extracts were determined using LC-MS analysis on an LC-30 AD (Shimadzu, Kyoto, Japan), equipped with a binary solvent delivery system and an auto-sampler, and coupled to the ESI source of a QTRAP 5500 Triple Quad mass spectrometer (AB SCIEX, California, USA). Separation was performed on an ACQUITY HSS T3 column (1.8 µm, 2.1 × 100 mm; Waters, Milford). The flow rate was 0.3 mL/min. The mobile phase consisted of (A) water and (B) acetonitrile, both of which contained 0.1% (vol/vol) formic acid. The optimized gradient elution program was performed as follows: 0.0–5.0 min: 10% B; 5.0–25.0 min: 10%–90% B; 25.0–30.0 min: 90% B; 30.0–30.1 min: 90%–10% B; 30.1–33.0 min: 10% B. The MS/MS spectrometry conditions included a capillary voltage of 4,500 V, a desolvation temperature of 450°C, and a desolvation gas flow of 360 L/h. The compounds were ionized in the positive ion mode (ESI+) for analysis. Precursor ion scanning signals were acquired from $m/z$ 50 to 500 Da, and product ions with a lactone ring were monitored at $m/z$ 102 Da. AHLs were identified and confirmed by comparing the retention times and the spectra with synthetic standards (C4-HSL, C6-HSL, 3OC6-HSL, C8-HSL, and 3OC8-HSL; Sigma). The standard curves were analyzed by multiple reaction monitoring (MRM) to assess relationships between the chromatographic retention times and mass spectra of AHL standards. The AHL content was measured by comparison with a calibration curve generated with each of the appropriate AHL synthetic standards.

## *N*-acyl-homoserine lactones add-back studies

AHLs add-back studies were carried out to further illustrate the regulation of different AHLs for amino acid metabolism. The AHLs detected in metabolomics and found by previous studies with a final concentration of 20 µM were supplemented to LB broth. Overnight cultures for Δ*luxI* were inoculated in LB broth with indicated AHLs. After incubation for 12 h, the levels of amino acid and related metabolic genes in Δ*luxI* with or without exogenous AHLs were determined based on the previously mentioned methods, respectively.

## RNA isolation and quantitative real-time RT-PCR

RNA samples were harvested from LB with or without exogenous AHLs at 12 h and subjected to total RNA extraction using a RNAprep pure Cell/Bacteria Kit (Tiangen Biotech, Beijing, China) according to the manufacturer's instructions. About 1,000 ng RNA was reversely transcribed into cDNA using a PrimeScript Reagent kit (RR047, Takara, Japan). Each 20 µL PCR mixture consisted of 1.6 µL template cDNA, 10 µL SYBR Premix Ex Taq (RR420, Takara, Japan), 0.4 µL each of the forward and reverse primers (10 mM), and 7.6 µL RNA free water. Amplification was performed with a Step-One Thermal Cycler (Applied Biosystems, United States) and consisted of 40 cycles of denaturation at 95°C for 15 s, annealing at 95°C for 30 s, and extension at 60°C for 45 s. The 16S rRNA gene was used as housekeeping control. The result was analyzed by the $2^{-\Delta\Delta CT}$ method (44).

## HPLC analysis of folates

The analysis of folates was conducted as described by Jastrebova et al. and Lin et al. (45, 46), with slight modification. Eight milliliters of the culture was added to 10 mL of extraction buffer (0.1 M phosphate buffer containing 0.5% sodium ascorbate). The mixture was placed in a boiling water bath for 15 min and then centrifuged at 4,000 rpm for 10 min. To 3 mL of supernatant, 550 µL of human plasma (Sigma, 15%, vol/vol) and 2-mercaptoethanol (Sigma) were added. The mixture was incubated at 37°C for 1 h under continuous rotation. The reaction was stopped by placing the samples in boiling water for 5 min. The extract was centrifuged at 20,000 rpm for 15 min. The supernatant was then filtered through a 0.22-µm filter and stored at −20°C until use. The chromatographic analysis was performed using an Agilent 1100 HPLC system. The C18 HypersilTM (25 × 4.6 mm, 5 µm) was used as the analytical column. The chromatographic conditions for gradient elution were as follows: flow-rate, 0.4 mL/min; volume injected, 20 µL; column temperature, 23°C; fluorescence detection, 290 nm excitation and 360 nm emission; UV detection, 290 nm. The mobile phase consisted of (A) a binary gradient mixture of 30 mM potassium phosphate buffer at pH 2.3 and (B) HPLC grade acetonitrile. The optimized gradient elution program was performed as follows: 0.0–5.0 min: 6% B; 5.0–20.0 min: 6–25% B; 20.0–35.0 min: 25% B; 35.0–45.0 min: 25–6% B; 45.0–50.0 min: 6% B.

## Effect of folic acid on the accumulation of glycine, serine, and methionine

Filtered folic acid solutions were added to LB culture medium at the indicated final concentrations of 12.5 µM, 25 µM, 50 µM, 100 µM, and 200 µM. The *H. alvei* WT and Δ*luxI* were cultured overnight in 100 mL LB. Cultures were inoculated to an optical density at $OD_{600}$ of 0.15 and grown for 12 h. The subsequent experiments will be carried out according to the methods of amino acid and qPCR.

### ACKNOWLEDGMENTS

This study was supported financially by the National Natural Science Foundation of China (no. 31871895).

Hou H., Yan C., Li X., and Zhang G. contributed to the study conception and design. Material preparation, data collection, and analysis were performed by Yan C. The first draft of the manuscript was written by Hou H. and Yan C. All authors commented on previous versions of the manuscript and approved the final manuscript.

### AUTHOR AFFILIATIONS

[1]School of Food Science and Technology, Dalian Polytechnic University, Dalian, China
[2]Liaoning Key Lab for Aquatic Processing Quality and Safety, Dalian, China

### AUTHOR ORCIDs

Congyang Yan  http://orcid.org/0000-0002-0612-3509

Hongman Hou 🔟 http://orcid.org/0000-0002-1283-9992

## FUNDING

| Funder | Grant(s) | Author(s) |
|---|---|---|
| MOST \| National Natural Science Foundation of China (NSFC) | No.31871895 | Hongman Hou |

## AUTHOR CONTRIBUTIONS

Congyang Yan, Conceptualization, Data curation, Formal analysis, Investigation, Methodology, Software, Validation, Visualization, Writing – original draft, Writing – review and editing | Xue Li, Conceptualization, Data curation, Software, Validation, Writing – review and editing | Gongliang Zhang, Conceptualization, Investigation, Methodology, Supervision, Writing – review and editing | Jingran Bi, Conceptualization, Validation, Writing – review and editing | Hongshun Hao, Conceptualization, Writing – review and editing | Hongman Hou, Conceptualization, Data curation, Formal analysis, Funding acquisition, Investigation, Methodology, Project administration, Supervision, Validation, Visualization, Writing – review and editing

## DATA AVAILABILITY

The raw and processed transcriptome data of *H. alvei* H4 have been deposited in the Gene Expression Omnibus (GEO) database under the accession number GSE137815.

## ADDITIONAL FILES

The following material is available online.

### Supplemental Material

**Supplemental material (Spectrum00687-23-s0001.docx).** Figures S1 to S4.

### Open Peer Review

**PEER REVIEW HISTORY (review-history.pdf).** An accounting of the reviewer comments and feedback.

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
