## [Reviewer comments · Microbiology Spectrum]

Microbiology Spectrum

AHL-Differential Quorum Sensing Regulation of Amino Acid Metabolism in *Hafnia alvei* H4

congyang yan, Xue Li, Gongliang Zhang, Jingran Bi, Hongshun Hao, and Hongman Hou

Corresponding Author(s): Hongman Hou, Dalian Polytechnic University

Review Timeline:

Submission Date:	February 14, 2023
Editorial Decision:	May 18, 2023
Revision Received:	July 20, 2023
Editorial Decision:	September 9, 2023
Revision Received:	November 8, 2023
Accepted:	January 20, 2024

Editor: Jannell Bazurto

Reviewer(s): Disclosure of reviewer identity is with reference to reviewer comments included in decision letter(s). The following individuals involved in review of your submission have agreed to reveal their identity: Jürgen Tomasch (Reviewer #2)

Transaction Report:

DOI: <https://doi.org/10.1128/spectrum.00687-23>

May 18, 2023

Prof. Hongman Hou
Dalian Polytechnic University
Dalian
China

Re: Spectrum00687-23 (AHL-Differential Quorum Sensing Regulation of Amino Acid Metabolism in *Hafnia alvei* H4)

Dear Prof. Hongman Hou:

Thank you for submitting your manuscript to Microbiology Spectrum. As you will see below, the reviewers recognized the interest and value in your work, both agree that substantial revision is required, experimentally and in the clarity of communicating the work. When submitting the revised version of your paper, please address all reviewer issues and provide (1) point-by-point responses to the issues raised by the reviewers as file type "Response to Reviewers," not in your cover letter, and (2) a PDF file that indicates the changes from the original submission (by highlighting or underlining the changes) as file type "Marked Up Manuscript - For Review Only". Please use this link to submit your revised manuscript - we strongly recommend that you submit your paper within the next 60 days or reach out to me. Detailed instructions on submitting your revised paper are below.

Link Not Available

Sincerely,

Jannell Bazurto

Journals Department
Reviewer comments:

Reviewer #1 (Public repository details (Required)):

Metabolomics and transcriptomics data included in this work, but I did not see links to a public repository.

Reviewer #1 (Comments for the Author):

Quorum sensing is an important process within bacteria for the coordination of physiological and metabolic processes with cell density. This work sets out to characterize a link between quorum sensing in *H. alvei*, a notable food spoilage microorganism, and amino acid metabolism, with a particular focus upon methionine, serine, and glycine. Overall, I think that this work begins

down the path of illustrating a connection between QS and amino acid metabolism and this characterization is useful to the field. However, as described below, I think there are some critical errors in the set-up and interpretation of some of the experiments (Figure 3) and also the authors might be missing possible connections to several key metabolic processes (folate metabolism). Throughout the work, the authors should also do a better job of describing specific pathways and processes (e.g. tautomerization cycle) and explaining the experimental approaches used with rationale for why this was the correct approach to answering that particular question (spot assays, transcriptomics. Etc.). The authors should also make sure to deposit all large datasets (metabolomics and transcriptomics) to a public repository, which I did not see links to in this version of the manuscript. English language polishing tools might also help with the general readability of the work.

General Comments:

1. The authors state in the introduction and show in Figure 1 that a mechanism of AHL biosynthesis involves the transfer of acyl groups onto SAM and the cleavage of the acyl-SAM intermediate to generate MTA and AHL products. This means that AHL biosynthesis relies upon flux toward methionine biosynthesis and conversion of methionine to SAM. Methionine is generated by MetE or MetH from homocysteine and 5-methyltetrahydrofolate (5-mTHF), where tetrahydrofolate (THF) is generated as a coproduct. Meanwhile, serine can be converted to glycine by serine hydroxymethyltransferase (GlyA), generating 5,10-methylenetetrahydrofolate (5,10-mTHF) from THF and glycine can be oxidized to CO₂ by the glycine cleavage complex (GCV), also generating 5,10-mTHF from THF. 5,10-mTHF is converted to 5-mTHF by MetF. Altogether, this means that THF metabolism forms a loop (THF → 5,10-THF → 5-THF → THF) and serine, glycine, and methionine metabolism all have a direct impact on the flux through this cycle.

Given this, it seems to me that AHL biosynthesis could be highly dependent upon folate metabolism and given the authors findings with glycine, serine, and methionine that folate metabolism links all these metabolites (much more than the author's suggestion in the introduction that the TCA cycle links these metabolites). Can the authors comment on this potential link between QS, folate metabolism, and the metabolism of methionine, serine, and glycine and how it affects the model suggested from their data? Additionally, can the authors depict in Figure 1 and include in the introduction a description of folate metabolism, as discussed above?

2. I think that the data presented in Figure 2 is difficult to quickly assess. I suggest that the authors migrate all of the data to Supplemental Materials Section and then only show panels for OD readings, and the 8/21 amino acids that were significantly affected by disruption of luxI (phenylalanine, tryptophan, serine, glycine, alanine, methionine, lysine, and proline) in Figure 2. Further, the authors should use panel designations 'A-I' to label each of the 9 panels presented in the new Figure 2. Please also describe the test and statistical cutoff used to designate statistically significant metabolite concentrations between WT and luxI mutants.

3. Generally, I do not think that the results from Figure 3 tell the authors anything about the effect of QS on amino acid metabolism. First, in the Methods section the authors state that they use YNB without amino acids or ammonium sulfate supplemented with glucose as minimal medium. However, if this is the case, WT and luxI mutant should show no growth, which is clearly not the case. Additionally, the Aspartate control (and YNB alone, which shouldn't have even grown) show growth discrepancies between the mutant and WT, generally. To me this indicates that contrary to the results in Figure 1, the WT and luxI mutant generally show different growth behavior across all minimal medium conditions. Therefore, the continued observance of a growth discrepancy when using different nitrogen sources does not really say anything about QS-mediated control of specific amino acid metabolism pathways. Overall, my assessment of Figure 3 is that the experiment was not properly controlled and therefore all results from panel A and panel B should not be considered in this work. Unless I am missing something, and the authors can explain how their cultures grew without nitrogen provided (YNB only) or the authors provide an alternative experiment that is properly controlled, I do not think that the conclusions made from these data are justified.

4. As described above, since I thought there were such critical issues with some of the results (namely, Figure 3), I have not reviewed the Discussion associated with this version of the manuscript. I am happy to review the remainder of the manuscript once the issues I have already described in my review are addressed in a subsequent revision of the manuscript.

Specific Comments:

Lines 48 & 50 - I am not sure whether 'interfere' is the correct word used in these instances. Please use a word like 'influences' or similar.

Lines 52-54 - This sentence is very clunky. Perhaps reword to something like "[...] and can also induce expression of yxB (encoding S-adenosylmethionine dependent methyltransferase) in *Bacillus subtilis* (11)."

Line 55 - I fail to understand how tautomerism or formation of different structural isomers, can effectively link these different amino acids, which are not structural isomers of each other. Can you please explain in more detail?

Line 60 and 61 - I think the statement that there 'must be a close' link between QS and amino acid metabolism is far to strong a statement to make in the introduction, given that the authors later state that studies have not looked in any great detail for such a link. Given the integrated nature of metabolism the authors have not presented any prior work assessing the 'closeness' of a QS-amino acid link and so I prefer that the authors reword to generally say that the above studies indicate a likely link between

QS and amino acid metabolism, but studies exploring this possible link are sparse.

Lines 61-63 - This sentence is also a bit clunky. Perhaps reword to say something like: "However, as far as the authors are aware, the only detailed study reporting a link between amino acid metabolism and QS is work detailing that amino acid-derived QS molecules can control the virulence of *Vibrio* species (14).

Line 65 - I am confused why the authors bring up hosts here.

Line 69 - If the QS system is involved in regulation of various pathogenic processes, please provide a more comprehensive list of these processes, more than just expression of extracellular proteases.

Line 71 - Can the authors clarify a bit further on what the STRING database is?

Line 72 - Amino acid metabolism is not one metabolic pathway but a large collection of (often integrated) pathways. Please be more specific as to which amino acid metabolism pathways were found to group with the QS system, according to the STRING analysis.

Line 81 - What is a self-metabolism pathway?

Line 87 - Please expand on what is meant by 'closely related'.

Line 92 - Biomass density and growth rate are different metrics for growth. Please reword, but I accept that deletion of *luxI* did not affect growth of the strain in LB medium, relative to WT.

Lines 103-104 - What does this mean? Do the authors mean to state that divergence between WT and the *luxI* mutant in concentrations for these particular metabolites all took place between the 6 and 9 h timepoints of the experiment?

Line 105 - The authors can eliminate the phrase 'reaching a certain cell density' and still convey the same message to the reader.

Line 110 - I think that the authors might mean to say that the minimal media contains one of the eight amino acids as a sole nitrogen source. As written, it sounds like all eight were mixed together for their test in this experiment.

Line 158 - Is this the same as the Pearson correlation coefficient? If so, please define what the *r* values are, both in the text and in Figure 4 legend.

Line 165 - Please describe in more detail how the transcriptomics analysis was performed here.

Lines 181-182 - I still don't understand what this tautomeric pathway is. Can the authors describe this exact pathway in more detail?

Lines 207-212 - These statements should include more references to the statistics associated with these findings. Seemingly, the addition of these compounds had no statistically significant influence on serine or methionine accumulation.

Reviewer #2 (Public repository details (Required)):

The RNAseq data as well as the metabolomic data need to be deposited in public databases.

Reviewer #2 (Comments for the Author):

In their manuscript „AHL-Differential Quorum Sensing Regulation of Amino Acid Metabolism in *Hafnia alvei* H4", the authors show that a knockout of autoinducer synthase gene results in an altered amino acid uptake dynamic compared to the wild-type. While the data is interesting, the presentation is overall very confusing and makes it hard to follow the reasoning of the authors. I think the manuscript needs substantial reorganization.

Most importantly: The genetic (and chemical) complementation assays are absolutely crucial for the experiments, thus the data in Fig 6 I and J belongs to Figure 2 and should be presented in a way that allows comparison of all amino acids for wild type, knock out and complemented strain. To me it seems that the genetic complementation could restore WT levels of not only glycine but also other amino acids, however this is difficult to judge from the current representation.

L116: Data on the composition of the minimal medium (modified YNB) is missing. It should be clearly defined. Actually, if it still contains yeast, it is still a complex medium with a nitrogen source present. I think these experiments need a much more detailed description.

Furthermore, while the overall conclusion that QS has a regulatory impact on amino acid biosynthesis pathways is supported by multiple evidence, I think the conclusion of a direct regulation of certain genes by LuxR as presented in Fig. 7 is too farfetched and should not be made based on the presented data.

The RNAseq data are very interesting however the number of replicates is not provided and also it seems that the authors did not deposit the raw data in a public database like GEO (same for metabolomic data).

While Fig. 1-3 are excellent and intuitively understandable, the other figures should be reorganized as they are too crowded, sometimes the text is barely readable. Consider presenting some of the data as supplementary figures and also keeping a minimal font size of 6pt.

Minor comments

L32: what is meant by "group-wide binding"? Odd wording

L37: AHLs can be much more diverse, e.g. the acyl side chain can be unsaturated in certain positions.

L50: "interfere" very imprecise wording.

L64: This sentence is redundant and not really needed here.

L76/77 Introduce abbreviations, the medium needs to be defined

L78: "linear relationship" is an odd choice here. You analyze the dependence of something based on the concentration of an effector molecule and may observe a linear relationship. You don't construct this relationship.

L81: What is meant by "self-metabolism"? Consider rewording.

L91: "grew ... during the growth curve" rephrase

L98: "which were enclosed in colored boxes" Descriptions of Figures do belong to the Figure legend not the main text.

L118: "was affected" imprecise wording

L146: I think this sentence belongs to the material and methods section

L165: Actually the relationship of AHL and Methionine biosynthesis pathway has already been described earlier in the ms.

L173: "it is obvious..." just state how many genes were up and down regulated.

L203: five signal-response QS systems: I don't agree with this description. You found 5 different molecules but only one synthase I guess. How many regulators of LuxR type?

L319: "QS can better participate" rephrase.

Overall the authors should consider streamlining the manuscript by avoiding redundant or overgeneralized sentences.

Staff Comments:

Preparing Revision Guidelines

Please return the manuscript within 60 days; if you cannot complete the modification within this time period, please contact me. If you do not wish to modify the manuscript and prefer to submit it to another journal, please notify me of your decision immediately so that the manuscript may be formally withdrawn from consideration by Microbiology Spectrum.

Response to Review 1 Comments

(Manuscript ID: Spectrum00687-23)

We thank the reviewer for the comments and suggestions to improve the manuscript.

1. The authors state in the introduction and show in Figure 1 that a mechanism of AHL biosynthesis involves the transfer of acyl groups onto SAM and the cleavage of the acyl-SAM intermediate to generate MTA and AHL products. This means that AHL biosynthesis relies upon flux toward methionine biosynthesis and conversion of methionine to SAM. Methionine is generated by MetE or MetH from homocysteine and 5-methyltetrahydrofolate (5-mTHF), where tetrahydrofolate (THF) is generated as a coproduct. Meanwhile, serine can be converted to glycine by serine hydroxymethyltransferase (GlyA), generating 5,10-methylenetetrahydrofolate (5,10-mTHF) from THF and glycine can be oxidized to CO₂ by the glycine cleavage complex (GCV), also generating 5,10-mTHF from THF. 5,10-mTHF is converted to 5-mTHF by MetF. Altogether, this means that THF metabolism forms a loop (THF -> 5,10-mTHF -> 5-mTHF -> THF) and serine, glycine, and methionine metabolism all have a direct impact on the flux through this cycle.

Given this, it seems to me that AHL biosynthesis could be highly dependent upon folate metabolism and given the authors findings with

glycine, serine, and methionine that folate metabolism links all these metabolites (much more than the author's suggestion in the introduction that the TCA cycle links these metabolites). Can the authors comment on this potential link between QS, folate metabolism, and the metabolism of methionine, serine, and glycine and how it affects the model suggested from their data? Additionally, can the authors depict in Figure 1 and include in the introduction a description of folate metabolism, as discussed above?

Your good advice is very much appreciated. **Firstly**, we have added and improved the content related to folate metabolism in Figure 1 and the introduction according to your suggestion (**revised manuscript, lines 56-62**). **Secondly**, we have explored the relationship between QS, folate metabolism, and the metabolism of methionine, serine, and glycine. Our study revealed that the QS system of *H. alvei* H4 reduced folate levels by promoting its degradation, while folate promoted the extracellular accumulation of glycine, serine, and methionine, indicating that the QS system indeed interferes with the metabolism of these three amino acids through the folate cycle (**revised manuscript, lines 206-236**). **Lastly**, as mentioned by the reviewer, we found that the addition of folate to LB medium can enhance the production of short-chain AHLs-type QS signals. Moreover, qRT-PCR experiments demonstrated that adding folate upregulated the expression of *luxI* gene encoding the QS signal synthesis

protein in *H. alvei* H4, which further confirmed the phenotypic results observed with the CV026 (revised manuscript, lines 237-253).

2. I think that the data presented in Figure 2 is difficult to quickly assess. I suggest that the authors migrate all of the data to Supplemental Materials Section and then only show panels for OD readings, and the 8/21 amino acids that were significantly affected by disruption of luxI (phenylalanine, tryptophan, serine, glycine, alanine, methionine, lysine, and proline) in Figure 2. Further, the authors should use panel designations 'A-I' to label each of the 9 panels presented in the new Figure 2. Please also describe the test and statistical cutoff used to designate statistically significant metabolite concentrations between WT and luxI mutants.

We agree with the suggestion of the reviewer and revised the Figure 2 based on your suggestion. The test and statistical cutoff used to designate statistically significant metabolite concentrations between WT and $\Delta luxI$ were described in the legend of Figure 2 (revised manuscript, lines 608-616).

3. Generally, I do not think that the results from Figure 3 tell the authors anything about the effect of QS on amino acid metabolism. First, in the Methods section the authors state that they use YNB without amino acids or ammonium sulfate supplemented with glucose as minimal medium. However, if this is the case, WT and luxI mutant should show no growth, which is clearly not the case. Additionally, the Aspartate control (and YNB

alone, which shouldn't have even grown) show growth discrepancies between the mutant and WT, generally. To me this indicates that contrary to the results in Figure 1, the WT and luxI mutant generally show different growth behavior across all minimal medium conditions. Therefore, the continued observance of a growth discrepancy when using different nitrogen sources does not really say anything about QS-mediated control of specific amino acid metabolism pathways. Overall, my assessment of Figure 3 is that the experiment was not properly controlled and therefore all results from panel A and panel B should not be considered in this work. Unless I am missing something, and the authors can explain how their cultures grew without nitrogen provided (YNB only) or the authors provide an alternative experiment that is properly controlled, I do not think that the conclusions made from these data are justified.

This experiment was carried out according to the method described by Schaefer et al. (1). The application of this method in the reference was also to find regulatory genes of the target metabolite. Based on your suggestion, we have reviewed the method, reagents used, and results of this experiment again, and found that there was an oversight in the description of the spot assay method, which resulted in some difficulty in interpreting the experimental outcomes. In fact, the YNB culture medium we used contains ammonium sulfate, which may be the reason why WT and $\Delta luxI$ can grow in only YNB. We sincerely apologize for this negligence and have repeated

the experiment, confirming that both bacteria can achieve a certain degree of growth in only YNB (containing ammonium sulfate).

In the YNB medium, aspartic acid (a non-QS-regulated amino acid) was added as a control for QS-regulated amino acids. In the solid phenotype results shown in Figure 3, there was no significant difference in the growth phenotype between the group with aspartic acid and the only YNB. It should be noted that the reference we consulted did not include the experiments conducted in liquid conditions, while the results shown in Panel A were intended to provide a more visual representation of the experimental results. However, the main analysis results were still based on the solid phenotype results in panel B. To avoid further complexity, we have removed the results from Panel A. We apologize for oversight in describing the components of YNB and we hope that our response can receive your support.

1. Schaefer K, Wagener J, Ames RM, Christou S, MacCallum DM, Bates S, Gow NAR. 2020. Three Related Enzymes in *Candida albicans* Achieve Arginine- and Agmatine-Dependent Metabolism That Is Essential for Growth and Fungal Virulence. *mBio* 11.

4. As described above, since I thought there were such critical issues with some of the results (namely, Figure 3), I have not reviewed the Discussion associated with this version of the manuscript. I am happy to review the remainder of the manuscript once the issues I have already described in my review are addressed in a subsequent revision of the manuscript.

According to your suggestion, we have clarified the issues regarding the experimental method and results of Figure 3 in recommendation 3. We

hope our response can explain the problems you raised. Thank you again for this very valuable suggestion.

Specific Comments:

5. Lines 48 & 50 - I am not sure whether 'interfere' is the correct word used in these instances. Please use a word like 'influences' or similar.

We agree with the comments of the reviewer and revised the word (revised manuscript, lines 53).

6. Lines 52-54 - This sentence is very clunky. Perhaps reword to something like "[...] and can also induce expression of yxbB (encoding S-adenosylmethionine dependent methyltransferase) in *Bacillus subtilis* (11)."

Thank you for your suggestion, but this part of content has been replaced by the folate cycle that you recommended (revised manuscript, lines 56-62).

7. Line 55 - I fail to understand how tautomerism or formation of different structural isomers, can effectively link these different amino acids, which are not structural isomers of each other. Can you please explain in more detail?

We agree with the comments of the reviewer. The manuscript wants to express the mutual transformation and influence of various amino acids through the TCA cycle. Using "automerism" in the manuscript was indeed inappropriate, and we have made the necessary modifications to the

manuscripts according to your suggestions (revised manuscript, lines 62-63, line 90, lines 165-166, line 184, line 199, line 310, line 324, line 633, respectively).

8. Line 60 and 61 - I think the statement that there 'must be a close' link between QS and amino acid metabolism is far too strong a statement to make in the introduction, given that the authors later state that studies have not looked in any great detail for such a link. Given the integrated nature of metabolism the authors have not presented any prior work assessing the 'closeness' of a QS-amino acid link and so I prefer that the authors reword to generally say that the above studies indicate a likely link between QS and amino acid metabolism, but studies exploring this possible link are sparse.

We thank the reviewer for this important suggestion and replace this sentence with “Altogether, the above studies indicate a likely link between QS and amino acid metabolism, but studies exploring this possible link are sparse” according to your suggestions. This really makes the interpretation more cautious (revised manuscript, lines 68-69).

9. Lines 61-63 - This sentence is also a bit clunky. Perhaps reword to say something like: "However, as far as the authors are aware, the only detailed study reporting a link between amino acid metabolism and QS is work detailing that amino acid-derived QS molecules can control the virulence of *Vibrio* species (14).

The suggestion provided by the reviewer was more appropriate for the manuscript and have better connectivity with the previous sentence. We have made modifications according to the reviewer's suggestions (revised manuscript, lines 70-72).

10. Line 65 - I am confused why the authors bring up hosts here.

The study of QS was not only limited to the information exchange and regulation between microorganisms, but also includes the relationship between QS system and host, which has been confirmed by some studies. Therefore, the complex relationship between the QS system and amino acid metabolism in pathogenic bacteria may have a certain impact on their host. However, the manuscript mainly explored the relationship between the QS system and amino acids metabolism in *H. alvei* H4, and the sudden occurrence of “hosts” may not match the overall situation. Therefore, we have removed the word “hosts”.

11. Line 69 - If the QS system is involved in regulation of various pathogenic processes, please provide a more comprehensive list of these processes, more than just expression of extracellular proteases.

In the revised manuscript, references about QS system involvement in spoilage and pathogenic regulation has been added as suggested by the reviewer (revised manuscript, lines 76-77).

12. Line 71 - Can the authors clarify a bit further on what the STRING database is?

STRING is a database of known and predicted protein-protein interactions. The interactions between proteins includes direct (physical) and indirect (functional) associations (revised manuscript, lines 373-374).

13. Line 72 - Amino acid metabolism is not one metabolic pathway but a large collection of (often integrated) pathways. Please be more specific as to which amino acid metabolism pathways were found to group with the QS system, according to the STRING analysis.

We have specifically pointed out the close relationship between the methionine metabolism pathway and the QS system, according to the STRING analysis (revised manuscript, lines 77-81).

14. Line 81 - What is a self-metabolism pathway?

The expression was not accurate. What we want to express was the regulation of different QS Signaling molecule on genes of glycine metabolism pathway. However, in the revised manuscript, research on the relationship between QS, folate cycle, and the metabolism of glycine, serine, and methionine has replaced this section.

15. Line 87 - Please expand on what is meant by 'closely related'.

The STRING database predicts the interactions between two proteins. We used this database to analyze the relationships between the QS gene cluster and other metabolic pathways including lipid metabolism, nucleotide metabolism, energy metabolism, metabolism of cofactors and vitamins, carbohydrate metabolism, and amino acid metabolism. It was found that

the QS gene cluster was most closely connected to the gene cluster associated with amino acid metabolism, suggesting a strong correlation between these two clusters. For more accurate expression, we have changed the term “closely related” to “closely connected” (revised manuscript, lines 95-97).

16. Line 92 - Biomass density and growth rate are different metrics for growth. Please reword, but I accept that deletion of luxI did not affect growth of the strain in LB medium, relative to WT.

We thank the reviewer for this important suggestion and revised this sentence according to your suggestion (revised manuscript, lines 105-106).

17. Lines 103-104 - What does this mean? Do the authors mean to state that divergence between WT and the luxI mutant in concentrations for these particular metabolites all took place between the 6 and 9 h timepoints of the experiment?

The sentence conveys the same meaning as the reviewer’s suggestion, and we have made the necessary modifications based on the reviewer’s suggestion (revised manuscript, lines 112-113).

18. Line 105 - The authors can eliminate the phrase 'reaching a certain cell density' and still convey the same message to the reader.

As suggested, the correction has been made (revised manuscript, line 113-116).

19. Line 110 - I think that the authors might mean to say that the minimal

media contains one of the eight amino acids as a sole nitrogen source. As written, it sounds like all eight were mixed together for their test in this experiment.

We are sorry for the unclarity and the issue has now been clarified in the revised manuscript (line 117-120).

20. Line 158 - Is this the same as the Pearson correlation coefficient? If so, please define what the r values are, both in the text and in Figure 4 legend.

As suggested, the Pearson correlation coefficient r has been defined in the text and Figure 4 legend (revised manuscript, lines 155-162 and lines 628-632, respectively).

21. Line 165 - Please describe in more detail how the transcriptomics analysis was performed here.

We cultivated WT and $\Delta luxI$ in LB medium for 12 h to obtain transcriptomic samples, and then selected amino acid metabolism-related genes with fold changes greater than 2 as target genes (revised manuscript, lines 174-176 and lines 177-178, respectively).

22. Lines 181-182 - I still don't understand what this tautomeric pathway is. Can the authors describe this exact pathway in more detail?

We apologize for the unclarity of this part and have replaced “tautomeric pathway” with “interconversion” (revised manuscript, lines 184).

23. Lines 207-212 - These statements should include more references to the statistics associated with these findings. Seemingly, the addition of

these compounds had no statistically significant influence on serine or methionine accumulation.

Indeed, almost all QS Signals have no significant effect on the accumulation of serine and methionine. In the revised manuscript, this section has been replaced by research on the relationship between QS system, folate cycle, and the metabolism of glycine, serine and methionine.

24. Metabolomics and transcriptomics data included in this work, but I did not see links to a public repository.

We have uploaded the raw data of the transcriptome to the Gene Expression Omnibus (GEO) database (www.ncbi.nlm.nih.gov/geo/) according to the suggestions of the reviewer (revised manuscript, lines 455-456). However, at present, the MetaboLights' website (www.ebi.ac.uk/metabolights/) cannot be opened normally in Chinese Mainland, resulting in the raw data of metabolomics not being uploaded to the public database, for which we sincerely apologize. we will continue to monitor the status of the database and provide the corresponding data as needed (corresponding author's email: houhongman@dlpu.edu.cn). Additionally, we have temporarily uploaded the entire metabolomics dataset to Baidu Netdisk (Link: https://pan.baidu.com/s/1qodiq9wK0hHEZhliH_UdfA, Extraction code: 1111), where the reviewers can download and verify the accuracy of the data.

Special thanks again to you for your good comments. It's really helpful for revising and improving our paper. All the lines and paragraphs indicated above are in the revised manuscript with red words.

Response to Review 2 Comments

(Manuscript ID: Spectrum00687-23)

We thank the reviewer for the comments and suggestions to improve the manuscript.

In their manuscript "AHL-Differential Quorum Sensing Regulation of Amino Acid Metabolism in *Hafnia alvei* H4", the authors show that a knockout of autoinducer synthase gene results in an altered amino acid uptake dynamic compared to the wild-type. While the data is interesting, the presentation is overall very confusing and makes it hard to follow the reasoning of the authors. I think the manuscript needs substantial reorganization.

1. Most importantly: The genetic (and chemical) complementation assays are absolutely crucial for the experiments, thus the data in Fig 6 I and J belongs to Figure 2 and should be presented in a way that allows comparison of all amino acids for wild type, knock out and complemented strain. To me it seems that the genetic complementation could restore WT levels of not only glycine but also other amino acids, however this is difficult to judge from the current representation.

We thank the reviewer for this important suggestion. Based on your suggestion, we have transferred Figures 6I and J to Figure 2 for presentation, and added metabolic comparisons of seven other QS-

regulated amino acids between WT, $\Delta luxI$, and $C-\Delta luxI$ strains in Figure 2. The results showed that the metabolic trends of all QS-regulated amino acids in the $C-\Delta luxI$ strain was almost consistent with that of the WT strain, indicating that these amino acids were regulated by the QS system (revised manuscript, lines 97-104 and lines 106-112, respectively).

2. L116: Data on the composition of the minimal medium (modified YNB) is missing. It should be clearly defined. Actually, if it still contains yeast, it is still a complex medium with a nitrogen source present. I think these experiments need a much more detailed description.

Special thanks to you for the comments. It is really true that we did not make it clear, and we have rewritten this part in the manuscript. The components of the minimal medium include 2% glucose, 0.17% YNB [without amino acids], and 1.5% agar. Among them, 100g of YNB medium included 5.0 g Ammonium Sulfate, 2.0 μ g Biotin, 400.0 μ g Calcium Pantothenate, 2.0 μ g Folic Acid, 2000.0 μ g Inositol, 400.0 μ g Niacin, 200.0 μ g p-Aminobenzoic Acid, 400.0 μ g Pyridoxine Hydrochloride, 200.0 μ g Riboflavin, 400.0 μ g Thiamine Hydrochloride, 500.0 μ g Boric Acid, 40.0 μ g Copper Sulfate, 100.0 μ g Potassium Iode, 200.0 μ g Ferric Chloride, 400.0 μ g Manganese Sulfate, 200.0 μ g Sodium Molybdate, 400.0 μ g Zinc Sulfate, 1.0 g Monopotassium Photosphate, 0.5 g Magnesium Sulfate, 0.1 g Sodium Chloride, 0.1 g Calcium Chloride (revised manuscript, lines 352-358).

3. Furthermore, while the overall conclusion that QS has a regulatory impact on amino acid biosynthesis pathways is supported by multiple evidence, I think the conclusion of a direct regulation of certain genes by LuxR as presented in Fig. 7 is too farfetched and should not be made based on the presented data.

As suggested, it is impossible to confirm whether dimers of the QS signal receptors and QS signal molecules directly regulates the expression of these genes based on presented data. Therefore, the statement in the manuscript that QS directly regulates these genes was not accurate, so in the revised manuscript, these regulatory descriptions have been changed to QS participating in the regulation of these genes (revised manuscript, line 289).

4. The RNAseq data are very interesting however the number of replicates is not provided and also it seems that the authors did not deposit the raw data in a public database like GEO (same for metabolomic data).

We apologize for the oversight and have added relevant information in the revised manuscript. Transcription transfer data includes 3 replicates of each experimental strain, while metabolomics includes 6 replicates of each experimental strain (**revised manuscript, line 379 and line 390, respectively**). In addition, we have uploaded the raw data of the transcriptome to the Gene Expression Omnibus (GEO) database (www.ncbi.nlm.nih.gov/geo/) according to the suggestions of the reviewer

(revised manuscript, lines 455-456). However, at present, the MetaboLights' website (www.ebi.ac.uk/metabolights/) cannot be opened normally in Chinese Mainland, resulting in the raw data of metabolomics not being uploaded to the public database, for which we sincerely apologize. But we will continue to monitor the status of the database and provide the corresponding data as needed (corresponding author's email: houhongman@dlpu.edu.cn). Additionally, we have temporarily uploaded the entire metabolomics dataset to Baidu Netdisk (Link: https://pan.baidu.com/s/1qodiq9wK0hHEZhliH_UdfA, Extraction code: 1111), where the reviewer can download and verify the accuracy of the data.

5. While Fig. 1-3 are excellent and intuitively understandable, the other figures should be reorganized as they are too crowded, sometimes the text is barely readable. Consider presenting some of the data as supplementary figures and also keeping a minimal font size of 6pt.

We greatly appreciate the reviewer's suggestions and have reorganized figures 4-7 in the revised manuscript, including presenting some data as supplementary materials and trying to maintain a minimum font size of 6pt. The modified figures have become more intuitive and easier to understand compared to the original ones.

Minor comments

6. L32: what is meant by "group-wide binding"? Odd wording

We agree with the comments of the reviewer and modify "group-wide binding" to "group-wide detection" (revised manuscript, lines 35-37).

7. L37: AHLs can be much more diverse, e.g. the acyl side chain can be unsaturated in certain positions.

As suggested, in describing the diversity of QS signaling molecule, we ignored the unsaturated condition of acyl side chain, and have modified it in the revised version according to the suggestions of reviewer (revised manuscript, lines 39-42).

8. L50: "interfere" very imprecise wording.

We agree with the comments of the reviewer and revised the word (revised manuscript, line 55).

9. L64: This sentence is redundant and not really needed here.

We have deleted the redundant sentence according to your suggestion.

10. L76/77 Introduce abbreviations, the medium needs to be defined

As suggested, we have rechecked all the abbreviations in the manuscript and made necessary modifications. Additionally, in the revised manuscript, we have defined the corresponding culture medium components in the methods section. (revised manuscript, lines 83-86, line 351, and lines 352-358, respectively).

11. L78: "linear relationship" is an odd choice here. You analyze the dependence of something based on the concentration of an effector molecule and may observe a linear relationship. You don't construct this

relationship.

We conducted a simple linear regression analysis between the five AHL-type QS signal molecules and three QS-regulated amino acids in *H. alvei* H4 grown in LB culture. As suggested by the reviewer, the statement was not precise enough. Therefore, we have replaced the “linear relationship” with “simple linear regression” to accurately reflect our analysis (revised manuscript, lines 86-89).

12. L81: What is meant by "self-metabolism"? Consider rewording.

The expression is not accurate. What we want to express is the regulation of different QS Signaling molecule on genes of glycine metabolism pathway. However, in the revised manuscript, research on the relationship between QS, folate cycle, and the metabolism of glycine, serine, and methionine has replaced this section.

13. L91: "grew ... during the growth curve" rephrase

As suggested, we have rewritten this sentence in the revised manuscript (revised manuscript, lines 104-105).

14. L98: "which were enclosed in colored boxes" Descriptions of Figures do belong to the Figure legend not the main text.

According to your suggestion, we have rearranged Figure 2, and consequently, the corresponding content has been removed.

15. L118: "was affected" imprecise wording

We changed the sentence to: “However, the growth of $\Delta luxI$ on solid

minimal medium with serine, methionine, and glycine individually added as the sole nitrogen source was significantly different from that of the WT strain.” (revised manuscript, lines 126-128).

16. L146: I think this sentence belongs to the material and methods section. As suggested, the section has been moved to the “MATERIALS AND METHODS” section (revised manuscript, lines 413-414).

17. L165: Actually the relationship of AHL and Methionine biosynthesis pathway has already been described earlier in the ms.

Indeed, in the "Introduction" section of this manuscript, we introduced the relationship between AHLs and the methionine biosynthetic pathway (revised manuscript, lines 41-48). However, this section mainly described the methionine metabolism related genes directly linked to QS system found through the PPIN of QS gene cluster and the amino acid metabolic gene clusters (revised manuscript, lines 168-171).

18. L173: "it is obvious..." just state how many genes were up and down regulated.

We agree with the comments of the reviewer and revised the sentence (revised manuscript, line 176).

19. L203: five signal-response QS systems: I don't agree with this description. You found 5 different molecules but only one synthase I guess. How many regulators of LuxR type?

We agree with the suggestion of the reviewer. Indeed, we only analyzed

the regulatory effects of five AHLs-type QS signal molecules and did not explore the homologous QS receptor protein. Therefore, we have changed “five signal-response QS systems” to “five QS signal molecules” in the manuscript (revised manuscript, line 165).

20. L319: "QS can better participate" rephrase.

Respectfully we would like to point out that this paragraph has been replaced by the discussion between QS system, folate cycle, and the metabolism of glycine, serine and methionine.

Special thanks again to you for your good comments. It's really helpful for revising and improving our paper. All the lines and paragraphs indicated above are in the revised manuscript with blue words.

September 9, 2023

Prof. Hongman Hou
Dalian Polytechnic University
Dalian
China

Re: Spectrum00687-23R1 (AHL-Differential Quorum Sensing Regulation of Amino Acid Metabolism in *Hafnia alvei* H4)

Dear Prof. Hongman Hou:

Thank you for submitting your manuscript to Microbiology Spectrum. As you will see below, the reviewers have reevaluated the revised manuscript and both agree that there is substantial improvement in the overall presentation of the data. However, there are still deficits in sufficiently communicating biological connections and drawing of some physiological conclusions. Text modification and additionally, minor experimental work suggested by reviewer 1 would greatly enhance the readability and impact of the manuscript and authors are welcome to resubmit a revised version of the manuscript for consideration.

Link Not Available

Sincerely,

Jannell Bazurto

Journals Department
Reviewer comments:

Reviewer #1 (Comments for the Author):

General Comments:

- 1) The abstract for this manuscript seems to no longer effectively capture the content presented in the edited manuscript. I recommend that the authors go back through the abstract and rewrite to better reflect their modified work.
- 2) I thank the authors for following up on my comment relating to tetrahydrofolate metabolism. Overall, I think that the authors have made a good effort to restructure the introduction in a way that follows much more. However, while I appreciate the authors' recognition of my writing skill by copying my text for Lines 56-63 and the legend of Figure 1, word for word, I would

prefer that the authors rewrite this section to reflect their voice and also include citations for all the statements made.

3) For lines 62-64, the manuscript that the authors cite is from work conducted in rice metabolism. I would prefer that the author cite work from *H. alvei* or at least a more closely related organism. Additionally, it is still not clear to me how the authors are suggesting the TCA cycle specifically links methionine, serine, and glycine metabolism. Please clarify the linkage between TCA and amino acid metabolism since it seems so critical to this work.

For instance, the authors cite work from *B. glumae* and QS impacting glyoxylate and oxalate metabolism as well as work from *Y. pestis* and impact of QS molecules on *gltA*, *sucCD*, and *sdhCBAB*. How are these assumed to influence glycine, serine, and/or methionine metabolism? I think that taking more time in the introduction to clearly draw links between TCA and specific amino acid metabolic pathways would really help set the groundwork for the rest of the manuscript.

4) The authors mention in their response to reviewers that the YNB medium contained amino acids. Therefore, the authors may not claim that any of these amino acids were tested in minimal medium using one of the eight amino acids 'as a sole nitrogen source', since other nitrogen sources are clearly present. However, they continue to make this claim in the 'Further screening and confirmation of amino acids regulated by QS' Section. If the authors insist upon keeping this section as written, they must perform a new experiment conducted with media that does not contain any other nitrogen source than the amino acids being tested. Far easier in my mind would be to rewrite the section to eliminate the claim that these experiments were conducted using the 8 amino acids as 'sole nitrogen sources' but I leave it to the authors to decide how they wish to move forward.

5) In figure 3, the authors use observations with aspartate as a sort of negative control. I am curious what the results look like for Figure 4E, if aspartate was used as a negative control in the same fashion as carried out for Figure 3. I would expect that the correlation of aspartate with the AHLs would be less than seen for Gly, Ser, and Met. Is this the case? Would the authors add these data to Figure 4E or as a Supplemental Figure to act as a control for this analysis?

6) Generally, the authors set out to explore Ser, Gly, and Met metabolism further, in the context of folate metabolism. It seems that the authors did not perform this in the context of a *luxI* mutant. That is, I would have been curious to see whether the addition of folate overcame (suppressed) any of the phenotypes seen for a *luxI* mutant, etc. Also are any of the results that the authors saw consistent with a feedback or feed-forward loop for QS biosynthesis? That is, it would be nice to see discussion about how these effects on Ser, Gly, and Met are predicted to impact biosynthesis of the QS molecule, generally.

7) Generally, there are still various grammatical mistakes scattered throughout the manuscript. I recommend that the authors take time to go back through the manuscript to fix these mistakes or have a third party go through the manuscript and correct, accordingly.

Specific Comments:

1) Line 23: I believe that the first use of *Hafnia alvei* in the abstract should be completely spelled out and not abbreviated to *H. alvei*.

2) Line 90: I do not believe that the term 'interconversion' is the most accurate word to use in this instance. Perhaps use 'metabolism' instead?

3) Line 102 and Figure 2: While I imagine that the amino acids measured are all 'L-' amino acids, the authors should denote this for each.

4) Lines 177-180: I am curious that if the authors took all the genes in the dataset, what fraction of the genes from the total dataset are identified as 'amino acid metabolism' genes? That is, is the 6.5% of hits from the transcriptomics dataset at all surprising (or enriched) given the fraction of genes in the genome denoted as being involved in amino acid metabolism, generally?

5) Lines 211-212: The authors claim that their data inform about the anabolism or biosynthesis of the chemicals in question. However, the data from Figure 6A only inform as to the relative abundance and not the mechanism by which a given metabolite accumulates or is depleted. Please reword accordingly.

6) Lines 237-244: The authors make mention of how the QS system upregulates expression of all the genes in Figure 7B. However, what the authors actually test is the expression of these genes upon the loss of the *luxI* system. Therefore, the authors should reword this section to denote that loss of *luxI* led to reduced expression of all genes in Figure 7B. However, somewhat importantly, this is not the same as showing that QS up-regulates expression of these genes.

7) Figure 3: Please denote the respective dilution in this figure for each column of spots.

8) Figure 6: How did the authors conduct the experiments described in Figure 6 G-I and J-O. Can the authors please add to the materials and methods the experimental setup for each of these experiments and add the raw data for G-I to a database, if this is from an RNA-seq experiment and the data is not already available?

Reviewer #2 (Comments for the Author):

The manuscript has been greatly improved, in particular the quality of the figures. The structure of the presented data is now more logical. Overall, the story that the authors want to tell is much better understandable now. However, I would propose some minor changes to the text to increase readability:

P1L15 "a common"

P2L35 delete redundant "a cell-density-dependent regulatory mechanism"

P3L48 delete "that has been extensively studied"

P3L65 Did you mean: can upregulate the biosynthesis of glyoxalate and oxalate branching from the TCA cycle?
P4L79 please describe in a few words what the STRING database contains, e.g. gene interactions derived from.... Not every reader is familiar with this database
P5L99 You detected metabolites in the supernatant fractions
P6L137 It is not exactly clear how biofilm formation relates to your data. I would delete the sentence and instead state that here you investigated responses to specific AHL molecules that you identified.
P7L157 I think the results of the statistical test obstruct the smooth reading of the sections. They can also be found in the figure and can therefore be deleted. Just state significance.
P7L163 Please be more specific here.
P8L179 please specify: Is it 6.5% of the regulated genes or 6.5% of the genes coding for amino acid metabolism
P9L211 on the folate metabolism
P11L253 Please be more specific here
P12L263 This study aimed at filling this gap by...
P12L271 Amino acids are not only important but essential
P12L272 Remove the redundancies from the sentence "Because microorganisms..."
P12L281 Clarify that you are talking about your own results here
P12L283 You don't need to specify the fold change cut-off here again.
P13L308 remove "mutual"
P15L343 G et al.?

Staff Comments:

Preparing Revision Guidelines

Please return the manuscript within 60 days; if you cannot complete the modification within this time period, please contact me. If you do not wish to modify the manuscript and prefer to submit it to another journal, please notify me of your decision immediately so that the manuscript may be formally withdrawn from consideration by Microbiology Spectrum.

Response to Review 1 Comments

(Manuscript ID: Spectrum00687-23R1)

We thank the reviewer for the comments and suggestions to improve the manuscript.

1. The abstract for this manuscript seems to no longer effectively capture the content presented in the edited manuscript. I recommend that the authors go back through the abstract and rewrite to better reflect their modified work.

Special thanks to you for the comments. It is really true that we did not make the conclusions clear, and we have rewritten the abstract in the manuscript (revised manuscript, lines 18-28).

2. I thank the authors for following up on my comment relating to tetrahydrofolate metabolism. Overall, I think that the authors have made a good effort to restructure the introduction in a way that follows much more. However, while I appreciate the authors' recognition of my writing skill by copying my text for Lines 56-63 and the legend of Figure 1, word for word, I would prefer that the authors rewrite this section to reflect their voice and also include citations for all the statements made.

We strongly believe that your description of the THF pathway is professional and concise, so we have directly adopted your description. But we have rewritten this section based on your suggestion (revised

manuscript, lines 60-64 and lines 604-606).

3. For lines 62-64, the manuscript that the authors cite is from work conducted in rice metabolism. I would prefer that the author cite work from *H. alvei* or at least a more closely related organism. Additionally, it is still not clear to me how the authors are suggesting the TCA cycle specifically links methionine, serine, and glycine metabolism. Please clarify the linkage between TCA and amino acid metabolism since it seems so critical to this work.

For instance, the authors cite work from *B. glumae* and QS impacting glyoxylate and oxalate metabolism as well as work from *Y. pestis* and impact of QS molecules on *gltA*, *sucCD*, and *sdhCBAB*. How are these assumed to influence glycine, serine, and/or methionine metabolism? I think that taking more time in the introduction to clearly draw links between TCA and specific amino acid metabolic pathways would really help set the groundwork for the rest of the manuscript.

We thank the reviewer for this important suggestion and have replaced this rice literature with microbiological literature. Additionally, some examples have been provided to illustrate how the TCA cycle affects the metabolism of glycine, serine, and methionine (revised manuscript, lines 66-72).

Respectfully we would like to point out that the inclusion of the two studies about *Yersinia pestis* and *Burkholderia glumae* in the subsequent sections was intended to demonstrate the relationship between the QS

system and the TCA cycle, rather than to illustrate the relationship between the TCA cycle and amino acid metabolism.

4. The authors mention in their response to reviewers that the YNB medium contained amino acids. Therefore, the authors may not claim that any of these amino acids were tested in minimal medium using one of the eight amino acids 'as a sole nitrogen source', since other nitrogen sources are clearly present. However, they continue to make this claim in the 'Further screening and confirmation of amino acids regulated by QS' Section. If the authors insist upon keeping this section as written, they must perform a new experiment conducted with media that does not contain any other nitrogen source than the amino acids being tested. Far easier in my mind would be to rewrite the section to eliminate the claim that these experiments were conducted using the 8 amino acids as 'sole nitrogen sources' but I leave it to the authors to decide how they wish to move forward.

Your good advice is very much appreciated. It is true that the YNB used in this study does contain ammonium sulfate, so it is inappropriate to state that the added amino acids were the only source of nitrogen in the experiments. We have rewritten this part in the manuscript according to the suggestions of reviewer (revised manuscript, line 129).

5. In figure 3, the authors use observations with aspartate as a sort of negative control. I am curious what the results look like for Figure 4E, if

aspartate was used as a negative control in the same fashion as carried out for Figure 3. I would expect that the correlation of aspartate with the AHLs would be less than seen for Gly, Ser, and Met. Is this the case? Would the authors add these data to Figure 4E or as a Supplemental Figure to act as a control for this analysis?

We thank the reviewer for this important suggestion and we have constructed a correlation between aspartate and QS signals according to the suggestions of the reviewers. These results were consistent with the reviewer's speculation that the Pearson r values of aspartate with the AHLs were all less than 0.88, and were shown in Figure S4 (revised manuscript, lines 164-167).

6. Generally, the authors set out to explore Ser, Gly, and Met metabolism further, in the context of folate metabolism. It seems that the authors did not perform this in the context of a *luxI* mutant. That is, I would have been curious to see whether the addition of folate overcame (suppressed) any of the phenotypes seen for a *luxI* mutant, etc. Also are any of the results that the authors saw consistent with a feedback or feed-forward loop for QS biosynthesis? That is, it would be nice to see discussion about how these effects on Ser, Gly, and Met are predicted to impact biosynthesis of the QS molecule, generally.

We agree and now performed an additional experiment to explore the effects of folate on the metabolism of glycine, serine, and methionine in

the $\Delta luxI$ strain. These results showed that the effects of folate on these three amino acids in the $\Delta luxI$ strain were weaker than that of WT strains, which provides evidence that folate enhances the regulatory role of the QS system in amino acid metabolism (revised manuscript, lines 220-223).

The QS system promotes the consumption of folate and the synthesis of THF, thereby accelerating the THF cycle. This situation should have promoted the conversion of homocysteine to methionine, leading to the accumulation of methionine. However, the QS system reduced the accumulation of methionine instead. These results suggest that the QS system was more inclined to promote the consumption of methionine, which was consistent with the positive regulation of *metK* expression by the QS system, ultimately enhancing the synthesis of QS signals (revised manuscript, lines 343-349).

7. Generally, there are still various grammatical mistakes scattered throughout the manuscript. I recommend that the authors take time to go back through the manuscript to fix these mistakes or have a third party go through the manuscript and correct, accordingly.

As suggested, we have checked and corrected the grammar issues throughout the manuscript.

8. Line 23: I believe that the first use of *Hafnia alvei* in the abstract should be completely spelled out and not abbreviated to *H. alvei*.

We are sorry for the oversight and the issue have now been clarified in the

revised manuscript (revised manuscript, line 17).

9. Line 90: I do not believe that the term 'interconversion' is the most accurate word to use in this instance. Perhaps use 'metabolism' instead?

We agree completely and we have replaced “interconversion” with “metabolism” based on your suggestion (revised manuscript, line 98, line 187, and line 220).

10. Line 102 and Figure 2: While I imagine that the amino acids measured are all 'L-' amino acids, the authors should demote this for each.

Thank you for your reminding. We have revised the question in the manuscript in lines 110-111, lines 117-120, and lines 133-134.

11. Lines 177-180: I am curious that if the authors took all the genes in the dataset, what fraction of the genes from the total dataset are identified as 'amino acid metabolism' genes? That is, is the 6.5% of hits from the transcriptomics dataset at all surprising (or enriched) given the fraction of genes in the genome denoted as being involved in amino acid metabolism, generally?

Through genome analysis of *H. alvei* H4, we identified 4406 genes in this strain, but we have not determined how many of these genes were related to amino acid metabolism. Transcriptome analysis between WT strains and $\Delta luxI$ revealed 946 significantly different genes. Among these genes, there are a total of 62 genes with differences greater than 2 and involved in amino acid metabolism, accounting for 6.5% of the 946 significantly different

genes.

12. Lines 211-212: The authors claim that their data inform about the anabolism or biosynthesis of the chemicals in question. However, the data from Figure 6A only inform as to the relative abundance and not the mechanism by which a given metabolite accumulates or is depleted. Please reword accordingly.

We thank the reviewer for the rigorous suggestion to improve the manuscript and we have performed corrections accordingly (revised manuscript, lines 212-214).

13. Lines 237-244: The authors make mention of how the QS system upregulates expression of all the genes in Figure 7B. However, what the authors actually test is the expression of these genes upon the loss of the luxI system. Therefore, the authors should reword this section to denote that loss of luxI led to reduced expression of all genes in Figure 7B. However, somewhat importantly, this is not the same as showing that QS up-regulates expression of these genes.

We agree with the suggestion of the reviewer and revised the sentence in the revised manuscript (revised manuscript, lines 245-246).

14. Figure 3: Please denote the respective dilution in this figure for each column of spots.

As suggested, we have annotated the respective dilution for each column of spots in figure 3.

15. Figure 6: How did the authors conduct the experiments described in Figure 6 G-I and J-O. Can the authors please add to the materials and methods the experimental setup for each of these experiments and add the raw data for G-I to a database, if this is from an RNA-seq experiment and the data is not already available?

As suggested, we added the method of the experiments described in Figure 6 G-I and J-O to the manuscript. The raw data for Figure 6 G-I was obtained through qPCR experiments, rather than from RNA-seq experiment (revised manuscript, lines 457-461).

Special thanks again to you for your good comments. It's really helpful for revising and improving our paper. All the lines and paragraphs indicated above are in the revised manuscript with red words.

Response to Review 2 Comments

(Manuscript ID: Spectrum00687-23R1)

We thank the reviewer for the comments and suggestions to improve the manuscript.

1. L15 "a common"

Special thanks to you for the comments. We have added the word according to your suggestion (revised manuscript, lines 15-16).

2. L35 delete redundant "a cell-density-dependent regulatory mechanism"

We agree completely and we have deleted the redundant sentence according to your suggestion (revised manuscript, lines 39-40).

3. L48 delete "that has been extensively studied"

As suggested, we have deleted the redundant sentence (revised manuscript, line 51).

4. L65 Did you mean: can upregulate the biosynthesis of glyoxalate and oxalate branching from the TCA cycle?

The intent of this paragraph was consistent with the reviewer's suggestions, and has been revised in the revised manuscript according to the reviewer's suggestions (revised manuscript, lines 72-74).

5. L79 please describe in a few words what the STRING database contains, e.g. gene interactions derived from.... Not every reader is familiar with this database

We apologize for the unclarity and have made appropriate changes in the lines 381-383 of the revised manuscript.

6. L99 You detected metabolites in the supernatant fractions

We agree completely and we have made the appropriate change (revised manuscript, lines 107-108).

7. L137 It is not exactly clear how biofilm formation relates to your data. I would delete the sentence and instead state that here you investigated responses to specific AHL molecules that you identified.

Special thanks to you for the comments. It is really true that this sentence is not relevant to this study, so we replaced it with “To investigate the roles of various QS signals in regulating amino acid metabolism, we first determined the question of how many different QS signaling pathways exist in *H. alvei* H4.” (revised manuscript, lines 145-146).

8. L157 I think the results of the statistical test obstruct the smooth reading of the sections. They can also be found in the figure and can therefore be deleted. Just state significance.

As suggested, we have rewritten this section in the revised manuscript (revised manuscript, lines 160-164).

9. L163 Please be more specific here.

Respectfully we would like to point out that Recommendation 8 has already conducted a detailed analysis of each QS signals and three target amino acid, so the sentence here has been deleted (revised manuscript, lines

167-168).

10. L179 please specify: Is it 6.5% of the regulated genes or 6.5% of the genes coding for amino acid metabolism

We are sorry for the unclarity and the issue have now been clarified in the revised manuscript (revised manuscript, lines 180-184).

11. L211 on the folate metabolism

As suggested, we have rewritten this section in the revised manuscript (revised manuscript, lines 211-212).

12. L253 Please be more specific here

We greatly appreciate the reviewer's suggestions and have rewritten this sentence (revised manuscript, lines 257-260).

13. L263 This study aimed at filling this gap by...

We agree with the suggestion of the reviewer and revised the sentence (revised manuscript, lines 267-269).

14. L271 Amino acids are not only important but essential

Following the reviewers suggestion we have revised the sentence in the revised manuscript (revised manuscript, lines 276-277).

15. L272 Remove the redundancies from the sentence "Because microorganisms..."

We agree completely and we have deleted the redundant sentence according to your suggestion (revised manuscript, line 277).

16. L281 Clarify that you are talking about your own results here

As suggested, we have clarified that the discussion here is based on our results (revised manuscript, line 284).

17. L283 You don't need to specify the fold change cut-off here again.

The sentence has been deleted following the comment of your suggestion (revised manuscript, lines 283-288).

18. L308 remove "mutual"

As suggested, we have deleted the redundant sentence according to your suggestion (revised manuscript, lines 309-310).

19. L343 G et al.?

We are sorry for the oversight and the issue has now been corrected in the revised manuscript (revised manuscript, lines 349-351).

Special thanks again to you for your good comments. It's really helpful for revising and improving our paper. All the lines and paragraphs indicated above are in the revised manuscript with blue words.

Re: Spectrum00687-23R2 (AHL-Differential Quorum Sensing Regulation of Amino Acid Metabolism in *Hafnia alvei* H4)

Dear Prof. Hongman Hou:

Thank you for your patience with the various rounds of revisions, your manuscript is much elevated and includes very exciting findings.

As your manuscript has been accepted, and I am forwarding it to the ASM production staff for publication. Your paper will first be checked to make sure all elements meet the technical requirements. ASM staff will contact you if anything needs to be revised before copyediting and production can begin. Otherwise, you will be notified when your proofs are ready to be viewed.

Sincerely,
Jannell Bazurto
Editor
Microbiology Spectrum